# Measurement report: Year-long chemical composition, optical properties, and sources of atmospheric aerosols in the northeastern Tibetan Plateau

**Kemei Li[1,3], Yanqing An[1], Jianzhong Xu[1,2*], Miao Zhong[1], Wenhui Zhao[1], Xiang Qin[1]**

[1]State Key Laboratory of Cryospheric Science and Frozen Soil Engineering, Northwest Institute of Eco-Environment and Resources, Chinese Academy of Sciences, Lanzhou 730000, China
[2]School of Oceanography, Shanghai Jiao Tong University, Shanghai 200030, China
[3]University of Chinese Academy of Sciences, Beijing 100049, China

Corresponding author: Jianzhong Xu (jzxu78@sjtu.edu.cn; jzxu@lzb.ac.cn)

## Abstract

Due to significant climatic effects, brown carbon (BrC) aerosol has received much attention in recent years. In this study, a year-long fine particular-matter ($PM_{2.5}$) samples were collected at Waliguan Baseline Observatory in the northeast of the Tibet Plateau to investigate the optical properties of water-soluble BrC and its source. The average concentration of $PM_{2.5}$ throughout the year was 10.3 ± 7.4 μg $m^{-3}$, with maximum in spring (14.0 ± 1.6 μg $m^{-3}$) and winter (12.5 ± 1.6 μg $m^{-3}$) and minimum in fall (7.95 ± 0.9 μg $m^{-3}$) and summer (7.14 ± 0.9 μg $m^{-3}$). Organic aerosol (OA) was the major component accounting for 37.7% on average, followed by sulfate (21.3%), nitrate (12.1%), and other species. OA and nitrate peaked during winter, while sulfate increased significantly during summer. Backward trajectory analysis on air mass reveals that the sources of the polluted air mass were mainly transported from the northeast and east of the sampling site. The seasonally average carbon-based mass absorption efficiency (MAE) of WS-BrC at 365nm were 0.92 ± 0.54 $m^2g^{-1}$ in spring, 0.40 ± 0.24 $m^2g^{-1}$ in summer, 0.81 ± 0.46 $m^2g^{-1}$ in fall, 0.97 ± 0.49 $m^2g^{-1}$ in winter, respectively. Comparison with other results, BrC in this study is weakly absorbed throughout the year, with that during the summer being the most photobleaching BrC. The chemical compositions of BrC are further investigated by parallel factorization analysis on the three-dimensional excitation-emission matrix and positive matrix factorization analysis on OA.

# 1 Introduction

Aerosols, tiny particulate matters suspended in the atmosphere, are critical climate forcing factor, such as on atmospheric radiation and the water cycle (Forster et al., 2021). Crucially, the influence of aerosols on climate is dictated by their physical and chemical properties, including mass concentration, number concentration, and chemical composition, which vary widely and unpredictably. This variability makes in-situ measurements essential for accurately assessing their impact. This is especially important for remote regions where aerosol loading is extremely low and strongly interacted with ambient conditions during transport. For example, during long-range transport, aerosols can interact with gas phase pollutants like nitrogen oxides ($NO_X$) and volatile organic compounds (VOCs), which can initiate photochemical reactions leading to the formation of secondary aerosols (Schnitzler and Abbatt, 2018; Fan et al., 2024). These transformations are further influenced by topographic and meteorological conditions that can mitigate the formation and evolution of aerosol characteristics including aerosols' optical properties (Schnitzler et al., 2022).

Aerosol optical properties are key parameters for evaluating their climatic effect. Brown Carbon (BrC) and Black Carbon (BC) represent two key optically sensitive components. BrC is particularly notable for its strong wavelength-dependent light absorption properties, distinct from the more uniform absorption characteristics of BC (Laskin et al., 2015). Originating from a variety of both anthropogenic and natural sources, BrC contributes significantly to the complexity of aerosol interactions within the atmosphere and remine the major uncertainty of aerosol light absorption estimation (Laskin et al., 2015; Yan et al., 2018).

The Tibetan Plateau (TP) is the highest and largest plateau on Earth. Aerosols in this receptor region mainly undergo long-range transport from the source region. Recent studies found that WS-BrC absorptivity in this remote and cold region has longer half-life than those in low altitude regions due to their lower decay-rate during transport (Choudhary et al., 2022). The higher aerosol loading and contribution of BrC on the TP mainly locates its margin due to the short distance from the source regions (Xu et al., 2024b). Qilian Mountains (QLM), situated on the northeastern margin of the TP represent a background region of inland of China. The importance of this region is represented by

its crucial hydrological resource for the arid northwestern region, which is essential for the sustenance of downstream communities and the ecological balance (Chen and Wang, 2009; Liu et al., 2017; Li et al., 2019). Precipitation in the mountain areas, through aerosol-cloud interaction, is the major origination (Qi et al., 2022). Xu et al. (2024a) emphasize the anthropogenic emission from the inland of China significantly increase the concentration of cloud condensation nuclei (CCN) in the QLM. However, the physical and chemical properties of aerosol in this background region is limited understood.

Research focusing aerosol in the QLM has been aroused increased interesting during last ten years (Che et al., 2011; Zhao et al., 2012; Zheng et al., 2015; Dai et al., 2021; Xie et al., 2022). It was found that inorganic components, especially for sulfate, accounted for a large proportion in the aerosol of QLM (Xu et al., 2014; Xu et al., 2015; Zhang et al., 2019; Zhang et al., 2020). Moreover, organic aerosol (OA) constitutes a significant fraction of the aerosol mass and exhibits significant chemical aging (Zhang et al., 2019; Zhang et al., 2020). Aerosol concentrations exhibit a notable seasonal variation. In spring, the QLM is predominantly affected by the prevalence of mineral dust, while during summer, the region experiences an influence of polluted air masses, which are conveyed from the northern and northeastern sectors of the TP (Xu et al., 2013). However, previous studies at the QLM are either short-term or discontinuous which are limited to represent the whole picture of aerosol properties in this region.

Located in the southeastern edge of the QLM, Waliguan Baseline Observatory (WLG) stands as a pivotal research site for understanding the atmospheric environmental variations both locally and regionally. To gain a deeper insight into the effects of human activities on aerosol in this region, this study conducted a year-long observation of aerosols at WLG to obtain the chemical composition, optical characteristics, seasonal variations, and sources of aerosols.

# 2 Sample collection and analysis

## 2.1 Sampling site

The WLG (36°17’ N, 100°54’ E; 3816 m a.s.l.) located at the top of the Waliguan Mountain in the northeastern TP, which is belong to the Global Atmosphere Watch (GAW) program of the World Meteorological Organization (WMO) (Figure 1). The Waliguan Mountain, with a relative elevation difference of about 600m above the ground (Figure 1b), is an ideal location for studying the background characteristics of atmospheric environment of inner Asia. The WLG is about 90km west of Xining, the capital of Qinghai Province with an elevation of ~2300 m a.s.l. The climate at this region is dominated by a distinct plateau continental climate, marked by pronounced Asian summer monsoon weather during summer and East Asian winter monsoon during winter. Spring and fall serve as transitional seasons between these two climatic systems.

## 2.2 Aerosol sampling

Fine particulate matter ($PM_{2.5}$) filter samples was collected on 47mm diameter quartz filter (PALL Life Sciences, USA) using a low flow aerosol sampler (Wuhan Tianhong Instrument Co. LTD, TH-16E) at a flow rate of 16.7 $L \cdot min^{-1}$. Before sampling, the quartz filters were baked in a Muffle oven at 550°C for 4h to remove the carbonaceous material. After the sampling, each filter was stored in a filter box packaged with clean aluminum foil. Subsequently, the box was saved in a ziplock bag and stored at –18°C. A total of 48 filter samples and three blank samples were collected during June 14, 2019 and May 6, 2020. Each sample was collected for 48h every seven days. The blank filters were obtained by being placed in the sampler for 10min without pumping. In this study, we divided the sampling period into different seasons as summer (6 June, 2019 to 28 August, 2019), fall (4 September, 2019 to 27 November, 2019), winter (11 December, 2019 to 26 February, 2020), and spring (4 March, 2020 to 6 May, 2020). The real-time meteorology data monitored by a Vantage Pro2 (Davis Instruments Corp., Hayward, CA, USA) weather station, including ambient temperature (T), relative humidity (RH), wind speed (WS) and wind direction (WD) were also obtained.

## 2.3 Chemical analysis

A 0.5 $cm^2$ piece of quartz filter was punched and used to determine organic carbon (OC) and elemental carbon (EC) content in $PM_{2.5}$. The rest of the filter was extracted by ultrasonication with 22 mL Milli-Q water (18.2 MΩcm) for 40 minutes and filtered by a 0.45μm PTFE filter (PALL Life Sciences, Ann Arbor, MI, USA). A suite of advanced instruments were employed to analyze the filtrate, including Ion Chromatography (IC) for water soluble ion speciation, Ultraviolet-Visible (UV-Vis) spectroscopy for absorbance spectrum of WSOC, Excitation-Emission Matrix (EEM) fluorescence spectroscopy for assessing fluorescence dissolved organic matter, Total Organic Carbon (TOC) analyzer for quantifying carbonaceous content, and offline analysis using a High-Resolution Time-of-Flight Aerosol Mass Spectrometer (HR-ToF-AMS) for detailed aerosol composition.

### 2.3.1 OC&EC

The OC/EC analysis was performed using a Thermal/optical Carbon Analyzer (DRI Model 2001; Desert Research Institute, Las Vegas, NV, USA) with the IMPROVE-A method (Chow et al., 2007). The 0.5 $cm^2$ quartz filter was loaded into the instrument, and then incrementally heated to 140°C (OC1), 280°C (OC2), 480°C (OC3) and 580°C (OC4), respectively, to vaporize OC. Then it was heated at 580°C (EC1), 740°C (EC2), 840°C (EC3) with oxidizing gas and 98% He/2% $O_2$ as carrier gas to vaporize EC. At each designated temperature stage, carbon is oxidized to $CO_2$ and then reduced to $CH_4$ by $H_2$ catalyzed by $MnO_2$. Ultimately, the hydrogen flame ionization detector is utilized to quantify the concentration of the resulting $CH_4$. In some samples, the EC concentration was lower than the minimum detection limit, and the EC content was not detected.

### 2.3.2 Ion chromatography

Eight water-soluble ionic species (WSIs) ($Na^+$, $NH_4^+$, $K^+$, $Ca^{2+}$, $Mg^{2+}$, $Cl^-$, $NO_3^-$, $SO_4^{2-}$) were determined using two 881 ion chromatography systems (Metrohm, Herisau, Switzerland). The cation system is facilitated by a Metrosep A Supp 5-250/4.0 column (Metrohm). The eluent composition consists of 3.2 mM $Na_2CO_3$ and 1.0 mM $NaHCO_3$, delivered at a flow rate of 0.7 mL

$min^{-1}$. The anionic chromatographic system is facilitated by a Metrosep C4-250/2.0 column. The mobile phase comprises 1.7 mM nitric acid and 0.7 mM dipicolinic acid (DPA), administered at a flow rate of 0.3 mL $min^{-1}$. To ensure optimal separation efficiency, the two columns temperature are maintained at 30°C, respectively. Prior to sample analysis, the instrument undergoes a rigorous calibration process. The IC measurements are determined by analyzing the retention time of the peaks and the integrated peak areas. These analytical parameters are then correlated with the calibration curve, which has been previously established using a series of standard solutions, to ensure accurate quantification of the analytes (Xu et al., 2015).

### 2.3.3 UV-vis

The ultraviolet-visible (UV-Vis) absorption spectrum of the samples was measured over a wavelength range of 200 nm–900 nm, with a resolution of 1nm, using a dual-beam UV spectrometer (UV-2700, Shimadzu, Kyoto, Japan). Samples were positioned in a quartz cuvette with an optical path length of 1cm and scanned at a rate of 5 nm $s^{-1}$, utilizing a dual light source system comprising deuterium and tungsten lamps. To correct baseline, the spectra of all samples were subtracted from the mean absorption value of the corresponding sample at 695 nm – 705 nm.

The absorption coefficient ($Abs_\lambda$) is calculated by Eq. (1) (Murphy et al., 2010).

$$Abs_\lambda = (A_\lambda - A_{700})\frac{V_l}{V_a \cdot l} \cdot \ln(10) \quad (1)$$

where $A_\lambda$ (M $m^{-1}$) is the absorption coefficient at a specific wavelength; $A_{700}$ is the mean absorption value at 695 nm–705 nm; $l$ is the light distance of the samples during the determination; $V_l$ is the volume of water used in extraction; $V_a$ is the volume of gas that passes through the quartz filter. In general, the absorption coefficient at wavelength 365 nm is used to refer to the absorption of brown carbon. The wavelength dependence of brown carbon absorption can be expressed by Eq. (2).

$$Abs_\lambda = K \cdot \lambda^{-AAE} \quad (2)$$

where $K$ is a constant related to aerosol mass concentration; $AAE$ is the absorption Ångström exponent of particulate matter, which is obtained by linear fitting the natural logarithm of the wavelength (300 nm–400 nm) to the natural logarithm of the corresponding $Abs_\lambda$. To calculate the

light absorption intensity of unit mass WSOC at a certain wavelength, the mass absorption cross section (MAE) is calculated by Eq. (3).

$$MAE = \frac{Abs_\lambda}{C_{WSOC}} \quad (3)$$

where $C_{WSOC}$ (μgC $m^{-3}$) is the concentration of water-soluble organic carbon in the atmosphere.

### 2.3.4 EEM

The three-dimensional excitation-emission matrix (3D-EEM) fluorescence of samples was scanned by an F-7100 fluorescence spectrometer (Hitachi High-Technologies, Tokyo, Japan), using 700-V xenon arc lamp as the excitation source. During the scanning process, the excitation (Ex) wavelengths ranged from 200 to 450 nm with intervals of 5 nm, while the emission (Em) wavelengths spanned from 250 to 600 nm with intervals of 1 nm. In this study, Milli-Q water (18.2 MΩ $cm^{-1}$) was used as a blank value for reference. Blank reference was subtracted from the EEM fluorescence spectra of the samples to eliminate the impacts of instrument to mitigate the instrumental effects. Once the EEMs were corrected, they were converted to Raman units (R.U.), after which the EEMs were subjected to modeling analysis (Murphy et al., 2013). Parallel factor analysis (PARAFAC), which is a three-way method, divides organic matter into different components based on the similarity of fluorescence characteristics. In addition to knowing the relative contribution of each component to the total fluorescence of organic matter, this method also provides information on the biochemical composition, origin, and biogeochemical action of the samples (Fellman et al., 2010). PARAFAC modeling was performed using DOMfluor and drEEM toolboxes installed on Matlab R2019a in this study. The entire process encompassed several stages: data preprocessing and preliminary analysis, followed by model construction and validation, culminating in the presentation of the final results (Stedmon and Bro, 2008).

In this study, humification index (HIX) and biological index (BIX) were used to analyze fluorescence spectral characteristics (Yang et al., 2020; Zhai et al., 2022). Notably, the HIX values exhibit disparities due to the distinct origins and transformation pathways of aerosol and aquatic samples. Therefore, the emission wavelength selected for HIX calculation was adjusted (from the commonly used 300-345nm to 325-365nm, and 435-480nm to 410-450nm) (Wen et al., 2021; Wu

et al., 2021). The above two optical indices can be calculated from Eq. (4) and Eq. (5), respectively (Zsolnay et al., 1999).

$$HIX = \frac{\sum SFI(410nm \le \lambda_{Em} \le 450nm)}{\sum SFI(325nm \le \lambda_{Em} \le 365nm)} (\lambda_{Ex} = 225nm) \quad (4)$$

$$BIX = \frac{SFI(\lambda_{Em} = 380nm)}{SFI(\lambda_{Em} = 430nm)} (\lambda_{Ex} = 310nm) \quad (5)$$

where $\lambda_{Em}$ is the emission wavelength; $\lambda_{Ex}$ is the excitation wavelength.

### 2.3.5 WSOC

Water-soluble organic carbon (WSOC) was measured by a total organic carbon analyzer (Elementar vario TOC cube, Hanau, Germany). The measurement was conducted by applying the total carbon (TC) and total inorganic carbon (TIC) method ($TOC = TC - TIC$). With oxygen as the carrier gas and platinum as the catalyst, inorganic carbon was transformed into $CO_2$ gas following acidification with 4% phosphoric acid. The concentration of $CO_2$ was determined using a non-infrared gas detector integrated within the instrument. Prior to the measurement, the total organic carbon (TOC) analyzer was calibrated with standard solutions of potassium hydrogen phthalate and sodium carbonate to ensure accurate quantification (Zhang et al., 2017).

### 2.3.6 HR-ToF-AMS off-line analysis and PMF source decomposition

The High-Resolution Time-of-flight Aerosol Mass Spectrometer (HR-ToF-AMS, Aerodyne Inc., Billerica, MA, USA) can obtain the information of chemical composition and particle size of non-refractory aerosol in real Time. HR-ToF-AMS mainly measures particles in the particle size range of 40–1000 nm. The instrument can not only observe aerosols online, but also analyze atomized aerosol extracts offline (Xu et al., 2015). Using argon as carrier gas, the samples were aerosolized and collected. The aerosol particles enter the HR-ToF-AMS through an aerodynamic lens, pass through a vacuum chamber and reach the hot surface at 600°C where they are vaporized instantly. Finally, it is bombarded with a 70 eV electron source and ionized into positively charged ion fragments, which enter the mass spectrometer for the detection of chemical components. According

to the different shapes of ion flight paths in the mass spectrum, HR-ToF-AMS has two operating modes, namely V mode and W mode. By comparing the data of W mode and V mode, we choose the data of V mode for the subsequent analysis. The data is processed using standard ToF-AMS data analysis software (Igor Pro 6.37). The software includes standard data processing toolkits SQUIRREL (v1.56) and PIKA (v1.15c). The processed matrix data were employed to investigate the sources of WSOA by positive matrix factorization (PMF). PMF source analysis is usually processed using the standard PMF evaluation tool (PET v2.03) developed based on Igor Pro software and the PMF2.exe algorithm (Ulbrich et al., 2009). Based on the Improved Ambient (I-A) method, relevant information of elemental analysis including oxygen-carbon ratio (O/C), hydrogen-carbon ratio (H/C), nitrogen-carbon ratio (N/C) and ratio of organic matter to organic carbon (OM/OC) can be obtained. The mass concentration of OM can be calculated using Eq. (6).

$$OM = OC \times (OM/OC) \tag{6}$$

where $OC$ is the mass concentration of OC measured by Thermal/optical Carbon Analyzer, $OM/OC$ is the ratio obtained from the above.

### 2.3.7 Backward Trajectory Model

To understand the possible source of the air mass during the sampling, the HYbrid Single-Particle Lagrangian Integrated Trajectory (HYSPLIT) model developed by the National Ocean and Atmospheric Administration (NOAA) and the Australian Bureau of Meteorology was used to calculate and analyze the backward transport trajectory of the air mass (Stein et al., 2015). The meteorological data used in this study are Global Data Assimilation System (GDAS) from the National Centers for Environmental Prediction (NECP), with 1°× 1° horizontal resolution. In the calculation, the height of observation station is set as 500 meters above the ground from the sampling site. Thereafter, hourly backward trajectories were performed for a duration of 72 hours to trace the air mass movements. Furthermore, the average backward trajectory cluster of the air mass during the sampling period was determined by assessing the spatial distribution similarities across all calculated trajectories.

The concentration-weighted trajectory (CWT) was used to analyze the source of pollution to the

sampling site. The CWT is a mixed-trajectory receptor model that combines meteorological trajectory nodes (residence time) and pollutant concentrations to trace their contributions to the pollution of a recipient site. After the study area was firstly gridded with a resolution of 0.25° × 0.25°, the CWT value of Grid (i, j) was calculated as follows:

$$CWT_{ij} = \frac{\sum_{l=1}^{M} C_l t_{ijl}}{\sum_{l=1}^{M} t_{ijl}} W_{ij} \tag{7}$$

$$W_{ij} = \begin{cases} 1.0(n_{ij} > 4n_{ave}); \\ 0.7(4n_{ave} > n_{ij} > n_{ave}); \\ 0.42(n_{ave} > n_{ij} > 0.5n_{ave}); \\ 0.05(n_{ij} < n_{ave}) \end{cases} \tag{8}$$

where $CWT_{ij}$ is the average weighted concentration in the cell $ij$; $M$ is the total number of trajectories; $C_l$ is the pollutant concentration when the trajectory $l$ through the grid $ij$; $t_{ijl}$ is the time that the trajectory $l$ stayed in the grid $ij$; $W_{ij}$ is the weight factor used to reduce the uncertainty of the calculation; $n_{ij}$ is the number of trajectory endpoints of grid $ij$, and $n_{ave}$ is the average number of trajectory endpoints. In this way, the CWT model is able to reveal regions that contribute significantly to the concentration of pollutants at the receptor site.

# 3 Results and discussion

During the sampling period, the meteorological conditions exhibited notable seasonal variations (Figure 2a). The average air temperature (±1σ) was 1.8 ± 8.3°C, with a daily maximum of 13.8 °C recorded on July 27, 2019, and a minimum of –15.8°C on December 26, 2019. Relative humidity (RH) ranged from 10% to 99%, with an average of 57 ± 28.1%. Seasonally, the average air temperatures were –2.7 ± 5.1°C in spring, 9.1 ± 3.5°C in summer, –1.6 ± 6.1°C in fall, and –10.0 ± 3.9°C in winter. Similarly, the average RH values were 47.0 ± 29.4% in spring, 88.3 ± 12.4% in summer, 68.3 ± 18.5% in fall, and 32.0 ± 16.1% in winter. Wind patterns were predominantly from the west during winter, with a step increase from the east during spring, reaching the predominance from the east in summer. Fall represented a transitional period (Figure 2a and 2b). Precipitation occurred primarily in summer (66.9%), followed by fall (17.2%) and spring (15.0%), with winter experiencing the least precipitation (0.9%).

## 3.1 Chemical speciation of $PM_{2.5}$

The total mass concentration of all species (WSIs + OM + EC) ranged from 2.0 μg $m^{-3}$ to 41.8 μg $m^{-3}$ during the study period, with a mean of 10.3 ± 7.4 μg $m^{-3}$ (Figure 2d). OM was the major contributor to aerosol mass concentration with an average contribution of 37.7%, followed by sulfate (21.3%), nitrate (12.1%), EC (1.1%), and other inorganic ions, which together accounted for 29.0% (including 7.5% $Na^{+}$, 7.6% $NH^{4+}$, 1.8% $K^{+}$, 6.7% $Ca^{2+}$, 0.8% $Mg^{2+}$, and 3.6% $Cl^{-}$) (Figure 2c). The mass concentrations were higher during spring (14.0 μg $m^{-3}$) and winter (12.5 μg $m^{-3}$), while relatively lower values were observed in summer (7.1 μg $m^{-3}$) and fall (8.0 μg $m^{-3}$) (Figure 2c). These seasonal patterns were driven by increased transport of polluted air masses from the east in winter and prevalent mineral dust storms in spring. The natural mineral dust reached its peak in spring (7.5% of $Ca^{2+}$) and its minimum in summer (4.1% of $Ca^{2+}$). The anthropogenic pollution markers ($SO_4^{2-}$ + $NO_3^{-}$) accounted for 33.2% of the mass in spring and 32.8% in winter. Among the secondary inorganic ions (sulfate, nitrate, and ammonium), sulfate was the most abundant, especially in summer, when its proportion reached 28.6%, similar to observations made by our group in July 2017 (Zhang et al., 2019). Sulfate formation during summer was mainly attributed to strong solar radiation, high humidity, and the heterogeneous reaction of $SO_2$ (Luo et al., 2022). In contrast, nitrate showed its minimum in summer (10.9%) and its maximum in winter (15.5%), which was mainly controlled by temperature-dependent partitioning. The average nitrate concentrations was 1.4 μg $m^{-3}$ with 2.0 μg $m^{-3}$ in spring, 0.8 μg $m^{-3}$ in summer, 0.9 μg $m^{-3}$ in fall, 1.9 μg $m^{-3}$ in winter in this study, which are comparable to measurements at WLG in July 2017 (0.7 μg $m^{-3}$) (Zhang et al., 2019) and at sites around the region, such as Qinghai Lake in the summer of 2010 (0.8 ± 0.5 μg $m^{-3}$) (Li et al., 2013) and Menyuan in autumn 2013 (1.7 μg $m^{-3}$) (Han et al., 2020). However, these concentrations are significantly higher than those recorded in the western Qilian Mountains, such as the summer 2012 observation at the Qilian Shan Station of Glaciology and Ecologic Environment (QSS) (0.6 μg $m^{-3}$) (Xu et al., 2015).

Ion balance, represented by the ratio of cation equivalent concentration (CE, neq $m^{-3}$) to anion equivalent concentration (AE, neq $m^{-3}$), was used to assess potential missing ions or the acid-base properties of aerosols (Xu et al., 2014; Xu et al., 2015). The CE/AE ratio calculated in this study

was 1.43 (Figure 3a), suggesting the potential presence of acidic aerosols, although carbonate and bicarbonate ions were not measured in the IC analysis. Assuming 2*[$HCO_3^-$] = [$Ca^{2+}$], the estimated CE/AE is still 1.35. In addition, the ratio of [$SO_4^{2-}$ + $NO_3^-$] to [$NH_4^+$] was 1.94, indicating that there was an excess of sulfuric and nitric acids. The acidic property in the aerosol of our study can be further supported by a significant number of organic acids, such as oxalic acid (Figure 3b). Oxalic acid is a product of atmospheric photochemical aging and is closely associated with sulfate and liquid water (Yang et al., 2009; Huang et al., 2019; Xu et al., 2020b; Boreddy et al., 2023). A moderate correlation was found between oxalic acid peak area and sulfate during summer ($R^2$=0.4) (Figure 3c).

Air mass backward trajectory analysis enables the initial tracing of potential sources and transport pathways of atmospheric aerosols throughout the observation period (Figure 4). Air mass origination varied from east to west seasonally, with the east mainly occurred during the summer transported with a shorter distance and the west during winter with a longer distance. Specifically, the fraction of the air mass from the east was up to 50.5% in spring and 66.0% in summer and the potential source areas for pollutants were predominantly associated with these air masses (Figure 5). The less important source areas are also observed from the north and west, especially during the fall, when the climatic systems of summer monsoon and the westerlies interacted. In these directions, widely distributed mineral dust source areas and sparse urban cities are located. Overall, anthropogenic emissions located in the east of WLG emerge as the most significant sources to the WLG.

## 3.2 Optical properties of WS-BrC

The average absorption coefficient ($Abs_{365}$) of WS-BrC at 365nm was 1.15 ± 0.97 $Mm^{-1}$. The $Abs_{365}$ was much higher in spring and winter than in summer and fall (1.55 ± 1.30 $Mm^{-1}$ in winter and 1.45 ± 0.54 $Mm^{-1}$ in spring vs. 0.88 ± 0.70 $Mm^{-1}$ in fall and 0.36 ± 0.21 $Mm^{-1}$ in summer), which is consistent with the distribution of OM mass concentration. The average absorption efficiency of WS-BrC at unit WSOC content (MAE) during the summer at 365nm ($MAE_{365}$, 0.40 ± 0.24 $m^2g^{-1}$) is significantly lower than that of the other three seasons (0.92 ± 0.54 $m^2g^{-1}$ in spring, 0.81 ± 0.46

$m^2g^{-1}$ in fall and 0.97 ± 0.49 $m^2g^{-1}$ in winter) (Figure 6a), suggesting highly photobleaching of BrC. $MAE_{365}$ in summer is comparable to that at WLG (0.48 $m^2g^{-1}$) in July 2017 (Xu et al., 2020a), Nam Co (0.38 $m^2g^{-1}$) from May 13 to July 1, 2015 (Zhang et al., 2017) and the regional background points of North China Plain (0.38 $m^2g^{-1}$) in summer of 2017 (Luo et al., 2020). But the $MAE_{365}$ in spring of this study (0.92 ± 0.54 $m^2g^{-1}$) is at a high level over the TP and even higher than the Qomolangma Station (QOMS) (0.81 $m^2g^{-1}$) which is frequently impacted by biomass burning emission (Xu et al., 2020a).

AAE of light absorption spectrum is an important optical parameter to check the containing of BrC in aerosols. In the 300–400 nm range, a high AAE value indicates significant aerosol absorption of shortwave ultraviolet light, with a relatively higher contribution from BrC. This phenomenon is typically observed in cases from biomass burning emission, secondary organic aerosols (SOA), and anthropogenic pollutant emissions (Siemens et al., 2022; Tao et al., 2024). The AAE (300 nm – 400 nm) in this study ranges from 3.06 to 8.42, with an annual average of 5.42 ± 1.26 peaking in summer at 6.21 ± 1.50, followed by 5.48 ± 0.96 in winter, 5.19 ± 1.00 in fall, and 5.14 ± 1.46 in spring (Figure 6a). The average annual AAE is comparable with the observation at Lulang (5.39 ± 1.22) during August 2014 to August 2015 at the southeastern TP (Li et al., 2016). The summertime AAE is similar to those at other stations in TP, such as Nam Co (5.91 ± 2.14) from May 13 to July 1, 2015 (Zhang et al., 2017) and WLG (5.96) from July 2017, but lower than those observed at QOMS (6.83) from April 12 to May 12, 2016 (Xu et al., 2020a).

Figure 6b illustrates the comparison of optical properties of WSOA in the map space of AAE (300 nm–400 nm) versus the logarithm of $MAE_{365}$ proposed by Saleh (2020). This map can be categorized into four classes as $MAE_{365}$ increase and $AAE_{300\text{-}400}$ decreases, which are associated with increased molecular sizes, decreased volatility, reduced solubility in water/organic solvents, and lower susceptibility to photobleaching. All our collected samples fall within the weakly absorbing BrC (W-BrC) category. This result is consistent with previous findings from QOMS and WLG reported by our group (Xu et al., 2022). In addition, the samples collected from other stations across the TP, including Nam Co station during summer (Zhang et al., 2017), Lulang station during

summer and winter (Wu et al., 2020), and Xining urban station during winter (Zhong et al., 2023), were also distributed in the W-BrC category. These results suggest that the samples at WLG during four seasons are aged BrC. Note that lower AAE and higher $MAE_{365}$ observed in spring were closer to moderately absorptive brown carbon (M-BrC) suggesting less oxidization.

## 3.3 Fluorescent components and fluorescence indices

PARAFAC analysis identify four components (C1-C4) in this study (Figure 7a). The chemical properties of each component are determined based on the comparison with previous studies (Chen et al., 2016a; Chen et al., 2016b; Chen et al., 2020; Yu et al., 2023; Zhong et al., 2023). C1 is determined as high-oxidation humus (HULIS-1) with the peaks of Ex and Em at 240 nm and 413 nm (Ex/Em = 240/413 nm) (Tang et al., 2024). C2 (Ex/Em = 225/375 nm) is classified as low-oxidation humus (HULIS-2), which is generally associated with combustion source (Li et al., 2022; Afsana et al., 2023). Both C3 (Ex/Em = 280 /358 nm) and C4 (Ex/Em = 225(270)/297 nm) were classified as protein-like organic matter (PLOM) (Wang et al., 2024). C3 is probably a fossil fuel-related substance (Wu et al., 2019) and C4 has a main peak and a secondary peak similar to the characteristics of tyrosine-like chromophore (Chen et al., 2016b; Chen et al., 2021b). HULIS compounds (C1 and C2) dominated the annual average contribution by 57.9%, of which C1 accounted for 22.9% and C2 accounted for 35.0%. PLOM contributed an average of 42.1%, with C4 accounting for 27.0% and C3 being 15.1% (Figure 7b). C1 presents a weak seasonal variation peaking in summer (23.54%) corresponding to the highest intensity of photochemical oxidation and contributing the least in spring (21.8%). The average relative contribution of C2 was 37.0%, 35.0%, 33.6% and 34.4% in spring, summer, fall and winter, respectively. The average relative contribution of C3 in spring (17.8%) and winter (17.0%) is higher than that in summer (12.6%) and fall (13.1%), which may be related to frequent coal-burning emissions during heating period. In contrast, the contribution of C4 is significantly more pronounced during summer (28.9%) and fall (30.5%) than that in spring (23.4%) and winter (25.1%), Corresponding to enhanced activities in agriculture and ecology (Zheng et al., 2016; Zhang et al., 2020).

An elevated degree of aging in WSOA is associated with an increased HIX value (Fan et al., 2020;

Wu et al., 2021; Ma et al., 2022) and a decreased BIX value (Wen et al., 2021). In this study, the average HIX and BIX values are 1.11 ± 0.18 and 1.29 ± 0.14, respectively, with seasonal variations of 1.04 ± 0.16 and 1.39 ± 0.24 in spring, 1.24 ± 0.11 and 1.26 ± 0.13 in summer, 1.13 ± 0.20 and 1.23 ± 0.09 in fall, and 1.02 ± 0.17 and 1.29 ± 0.09 in winter. The spring samples exhibit the greatest variability, indicating their fresher properties (Figure 8b). Summer season is characterized by the highest HIX and the low BIX, suggesting a high degree of aging and oxidation of WS-BrC. These values are positioned in the upper left corner of HIX versus BIX space (Figure 8a). Comparing with the results of previous study, the properties of aerosols in this study are more consistent with those in the northwestern China (Figure 8) (Chen et al., 2021a; Zhang et al., 2021a; Zhong et al., 2023), which is less humified than that in the eastern China.

## 3.4 Chemical components of WSOA and their absorption

PMF decomposes the WSOA into two factors, i.e., a more oxidized oxygenated OA (MO-OOA) and a less oxidized oxygenated OA (LO-OOA) (Figure 9a). The spectra of these two OOAs in this study are consistent with those of online measurement at Nam Co Station in the TP during the summer (Xu et al., 2018). The average mass contribution of LO-OOA and MO-OOA were 47% and 53% (Figure 9c), respectively. The mass contribution of MO-OOA across the four seasons (spring to winter) was 55.4%, 54.9%, 61.7% and 42.0%, respectively. The time series of LO-OOA correlated well with nitrate ($R^2$=0.47) during winter and less well with sulfate ($R^2$=0.39), while MO-OOA correlated poorly with sulfate and nitrate (Figure 9b).

The triangle plot of *m/z* 44 (*f*44) versus *m/z* 43 (*f*43) and Van Krevelen diagram of elemental ratios are valuable tools for examining ambient evolution of OA (Ng et al., 2010; Zhang et al., 2019; Chazeau et al., 2022). *f*44 is associated highly with oxidized oxygenated OA, while *f*43 corresponds to less oxidized OA. During oxidation, OA transited from a lower to a higher oxidation state, characterized by an increase in *f*44 and a decrease in *f*43, moving from the base to the apex of the triangular plot (Flores et al., 2014). Most data points in our study locate in the upper section of this triangular (Figure 9e) with data points during winter at a lower position and data points during summer shifting towards higher position, presenting a distinct oxidation degree at different seasons.

The Van Krevelen plot further elucidates the chemical transformations of OA during atmospheric aging (Heald et al., 2010; Xu et al., 2018). The transition slope from low to high oxygen states typically ranges from –1 to –0.5 (Ng et al., 2011). In our data, the linear regression slope of all data points is –0.62 (Figure 9f), higher than winter's –0.89 at Xining and summer's –0.76 at NamCo (Xu et al., 2018; Zhong et al., 2023). Seasonal slopes vary, with spring and summer at –0.58, fall at –0.60, and winter at –0.66, indicating different OA oxidation pathways during each season.

The light absorption characteristics of different WSOA factors, were evaluated by a multiple linear regression (MLR) model to assign the WSOA factors to the $Abs_{365}$ (Zhang et al., 2021b; Jiang et al., 2023). The MLR method can be expressed as Eq. (9).

$$Abs_{\lambda} = f_1 \times C_{MO-OOA} + f_2 \times C_{LO-OOA} \quad (9)$$

where $f_n$ is the corresponding fitting coefficients, which can also represent the mass absorption cross section (MAC) values of different organic components; $C_{MO-OOA}$ and $C_{LO-OOA}$ (μg $m^{-3}$) are the mass concentration of the organic components; $f \times C$ is the absorption value of the organic component. The $MAC_{365}$ of the two factors are 0.41 $m^2g^{-1}$ (MO-OOA) and 0.45 $m^2g^{-1}$ (LO-OOA) (Figure 6a). The $MAC_{365}$ value of LO-OOA is slightly higher than that of MO-OOA, which is related to the relatively weak photobleaching of LO-OOA. Compared to previous studies, $MAC_{365,\ MO-OOA}$ in this study is lower than $MAC_{370,\ MO-OOA}$ (0.60 $m^2g^{-1}$) at the QOMS (Zhang et al., 2021b). $MAC_{365,\ LO-OOA}$ is much lower than that observed in urban stations of Northwest China in winter 2019 (1.33 $m^2g^{-1}$) (Zhong et al., 2023), which may be attributed to strong photobleaching of OA in remote areas during atmospheric transport.

## 3.5 Relationship between oxidation state and optical properties of BrC

During the aging process of BrC, changes in its optical properties can reflect alterations in its chemical characteristics (Alang and Aggarwal, 2024). In this study, we investigated the relationship between $MAE_{365}$ and the elemental ratios of O/C and H/C across different seasons (Figure10). $MAE_{365}$ exhibited a positive correlation with O/C in spring ($r$ = 0.63; P < 0.01), while an insignificant negative correlation was observed in summer and fall ($r$ = 0.29 and $r$ = 0.09).

Conversely, the relationship between $MAE_{365}$ and H/C showed an opposite pattern in each season (Figure 10b). These results suggest that the light absorption capacity of BrC was enhanced during the oxidation process in spring due to functionalization or oligomerization, while further oxidation in summer and autumn leads to the fragmentation of large molecular weight compounds, resulting photobleaching, which diminishes the light absorption capacity (Jiang et al., 2022).

Furthermore, the optical evolution of WS-BrC during the oxidation process was explored by integrating the PARAFAC components with the WSOA components in EEM plot (Figure 10c). The compounds are categorized based on their correlation analysis among each other: C1 is strongly associated with MO-OOA, whereas C2 and C3 are linked to LO-OOA, and C4 exhibits a weak correlation with these two factors. Through this method, the chemical evolution of different components could be cross-validated and provides additional insights in the plot (Chen et al., 2016b; Zhong et al., 2023). Simply, the transition of less oxidized to highly oxidized OA through photochemical reactions can be applied to the process of BrC. Correspondingly, the optical evolution of BrC can serve as evidence of the oxidative state transition. For our dataset, C1 is likely formed through atmospheric oxidation processes similar to the transition from LO-OOA to MO-OOA, whereas C2 and C3 may originated from primary DOM in less oxidized region. C4 is the protein-like compound and has weak connection with OOA species.

# 4 Conclusions

In this study, atmospheric aerosol samples collected at WLG were analyzed, focusing on their chemical composition, optical properties, and sources. The main conclusions are as follows:

OM is the largest component of $PM_{2.5}$, accounting for an average of 37.7% of the mass, followed by sulfate (21.3%) and nitrate (12.1%). Notably, during summer, atmospheric photochemical reactions lead to significant sulfate production. The light absorption capacity of WS-BrC varies seasonally, with the highest levels observed in winter, followed by spring, fall, and summer. In summer, the AAE, and HIX are elevated, likely due to increased oxidation processing of OA. The

sources of aerosol to WLG are predominantly from the eastern urban areas.

Four chromophores are identified based on PARAFAC analysis, with HULIS being the predominant contributors to fluorescence. PMF analysis on OA revealed two factors of MO-OOA and LO-OOA. On average, MO-OOA is more dominant in mass concentration; however, its light absorption capacity is lower than that of LO-OOA. Both factors exhibit reduced light absorption compared to those in urban studies, indicating a high level of photochemical oxidation at WLG.

Overall, this study provides valuable insights and serves as a foundational reference for future research on atmospheric aerosol conditions in the northeastern Tibetan Plateau. The findings will aid efforts to better understand the background characteristics of aerosols in this region.

# Data availability

The data used in this study can be accessible at National Cryosphere Desert Data Center (https://www.doi.org/10.12072/ncdc.nieer.db6809.2025).

# Author contributions

JX designed the research and KL, MZ, and WZ collected samples. KL and JX processed data, plotted the figures, and wrote the manuscript when JX and MZ gave constructive discussion. YA and XQ had an active role in supporting the experimental work. All authors contributed to the discussions of the results and refinement of the manuscript.

# Competing interests

The authors declare that they have no conflict of interests.

# Acknowledgment

This study was supported by grants from the National Natural Science Foundation of China

(42476249 and 42021001), and the Fundamental Research Funds for the Central Universities. Thanks for the logistic support and assistance from WLG station.

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

# Figure

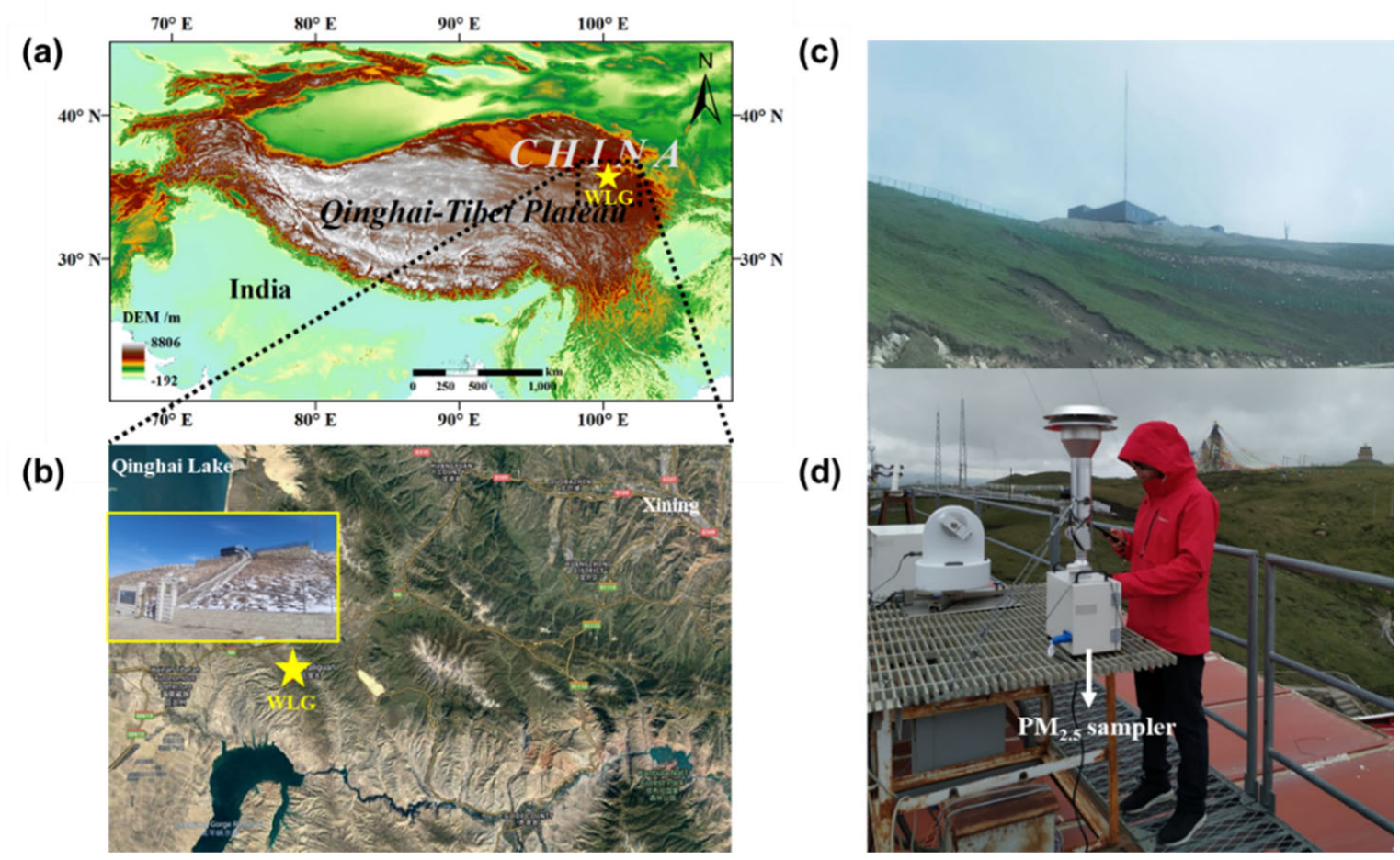

Figure 1 (a, b) Location map of the Waliguan Baseline Observatory on the TP, adapted from Zhao et al. (2022) (© Google Maps 2025). (c, d) Photographs of Waliguan Baseline Observatory and *in-situ* $PM_{2.5}$ sampling.

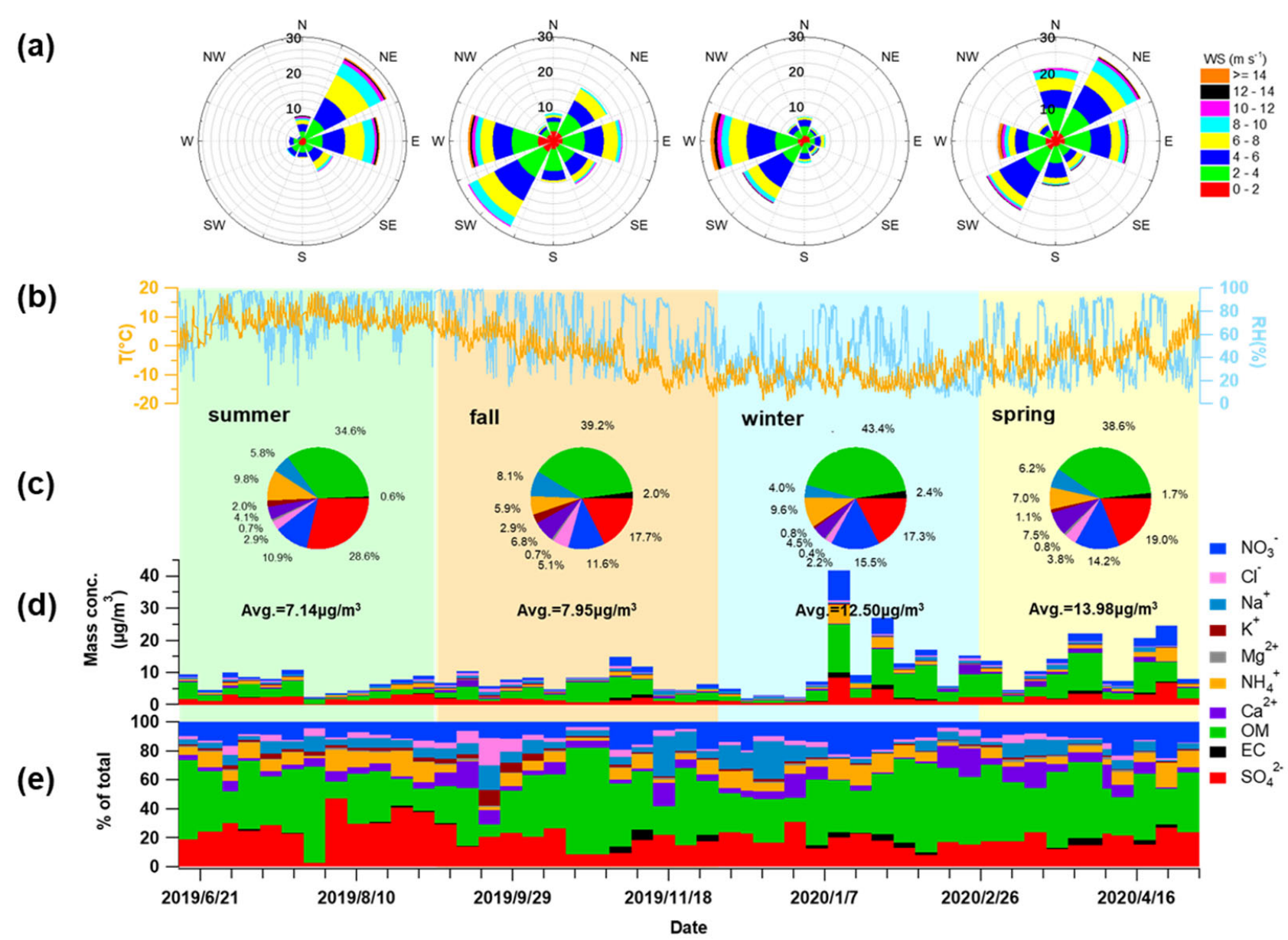

Figure 2 (a) Wind-rose diagram for four seasons, (b) time series of air temperature (T) and relative humidity (RH), (c) average chemical composition for four seasons, (d) mass concentration of all

species (WSIs + OM + EC), and (e) percentage of total mass concentration by species.

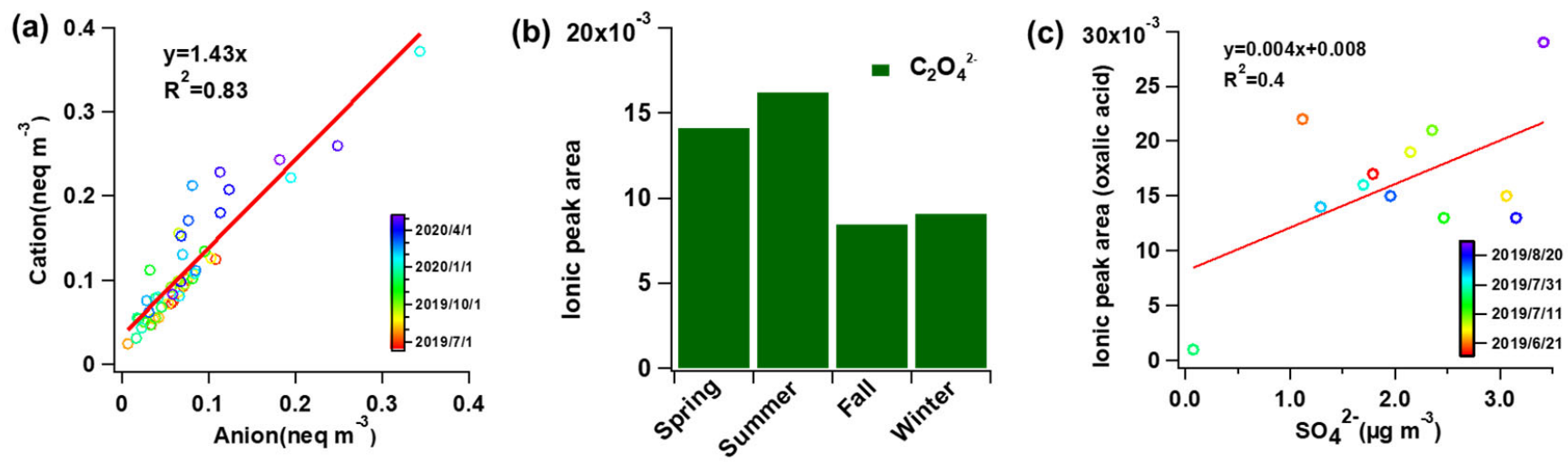

Figure 3 (a) The charge balance between the cations ($Na^+ + NH_4^+ + K^+ + Ca^{2+} + Mg^{2+}$) and anions ($Cl^- + NO_3^- + SO_4^{2-}$). (b) Ion peak areas of oxalic acid for four seasons, and (c) correlation between oxalate ion peak area and mass concentration of sulfate during summer.

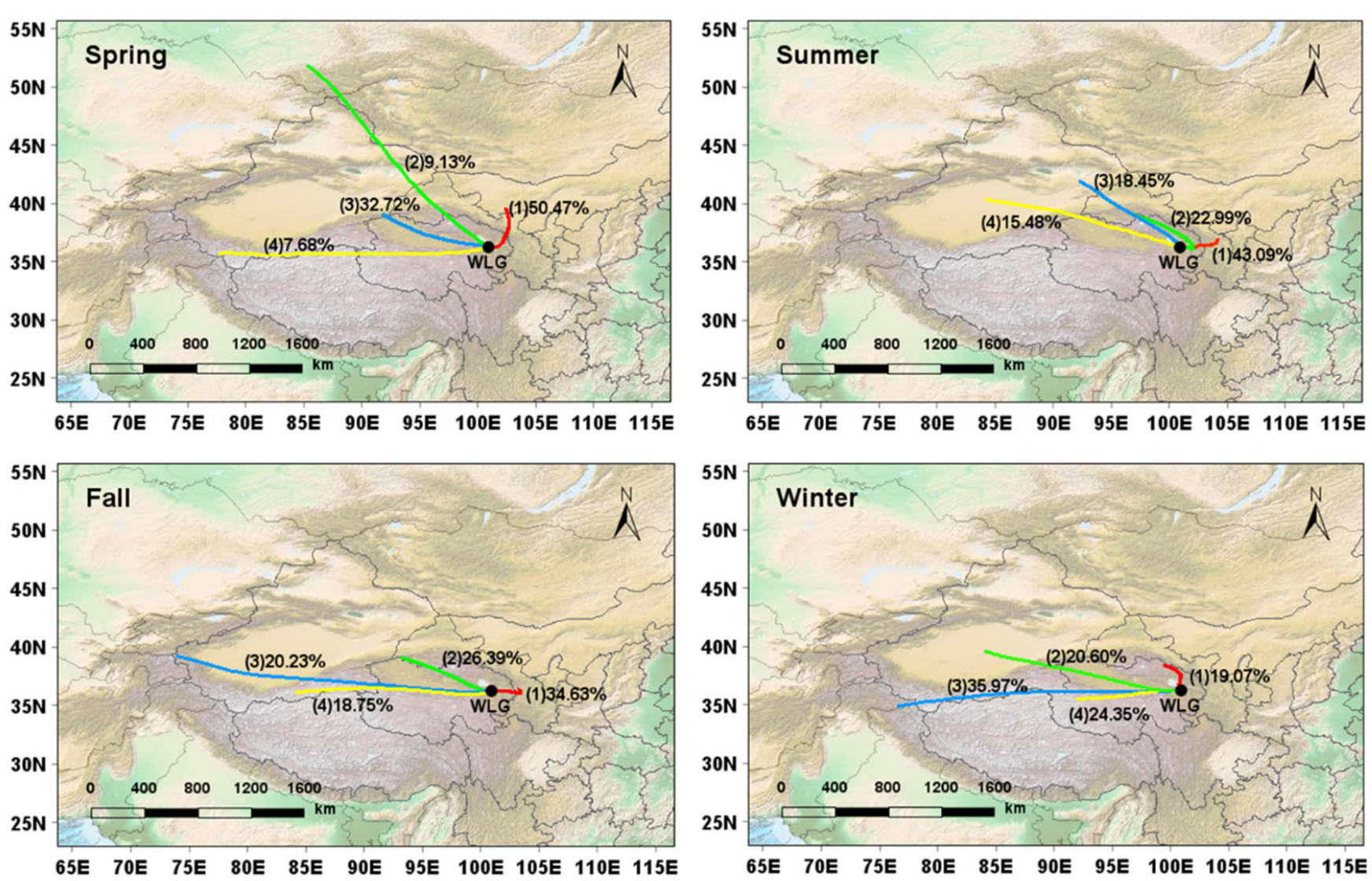

Figure 4 The average backward trajectory clusters and percentage of air mass for the four seasons during the observation period. The map is plotted in MeteoInfo 3.6.0.

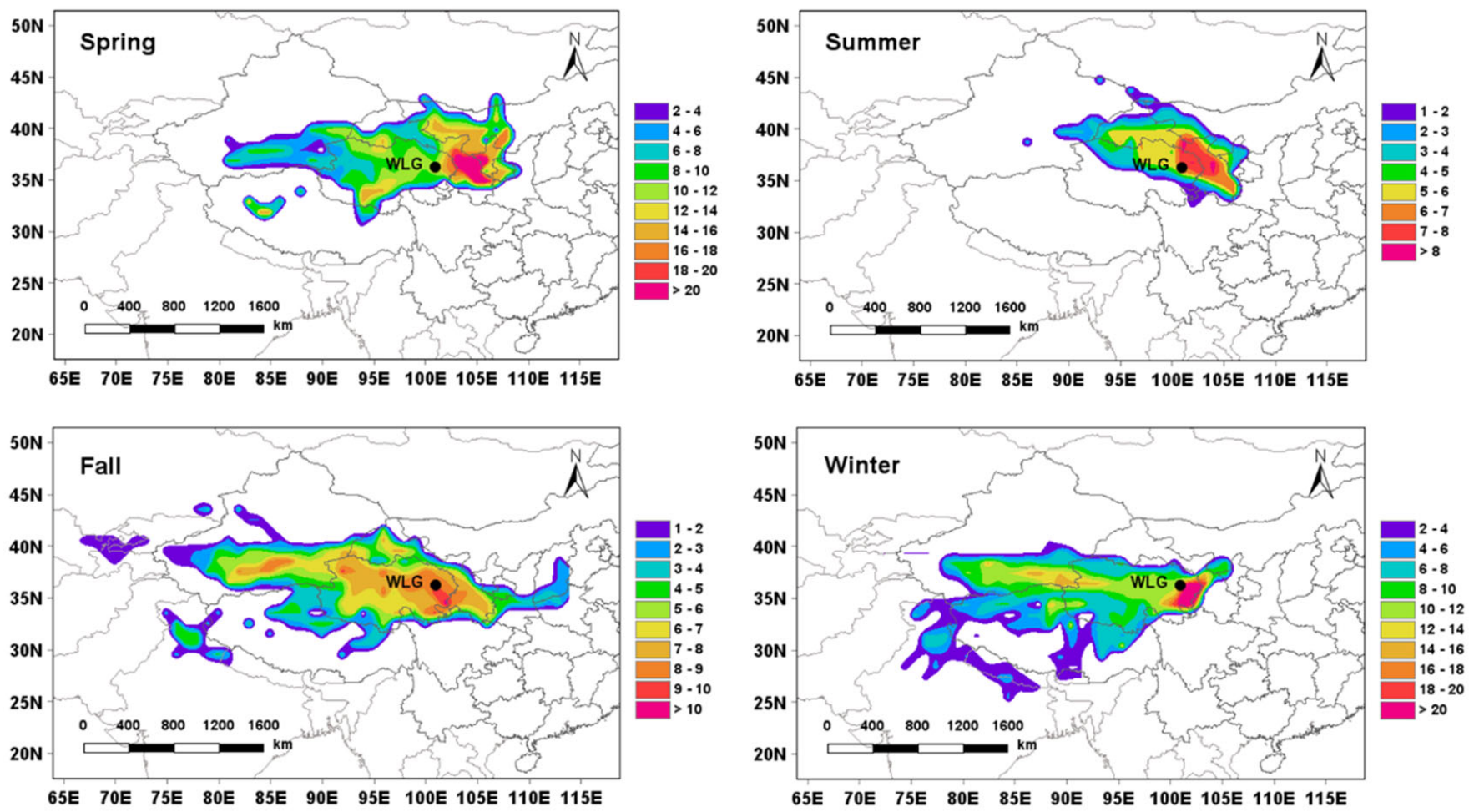

Figure 5 Map of CWT analysis of WLG $PM_{2.5}$ in four seasons plotted in MeteoInfo 3.6.0.

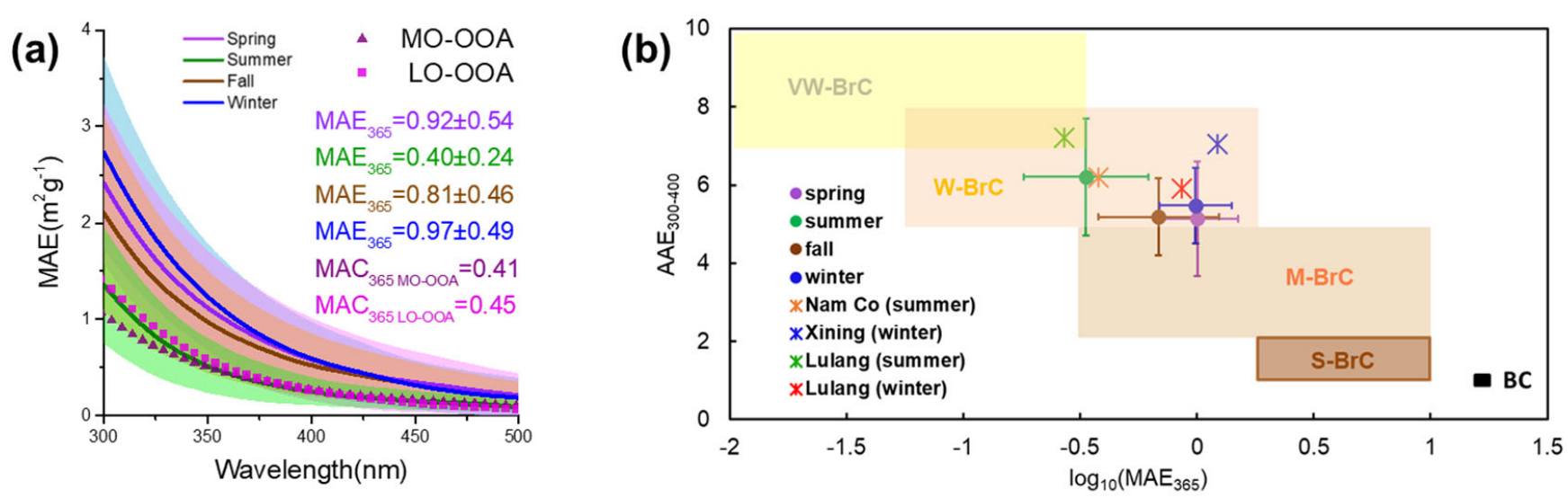

Figure 6 (a) The average MAE spectrum and standard deviations of WS-BrC in different seasons and two factors of WSOA analysed by PMF. (b) Optical-based BrC classification map in AAE-$\log_{10}(MAE_{365})$ proposed by Saleh (2020). The shaded squares in the map from left to right represent "very weakly" (VW), "weakly" (W), "moderately" (M), and "strongly" (S) absorbing BrC classes and black carbon (BC). The irregular marks are different station data from TP that have been reported by other researchers before.

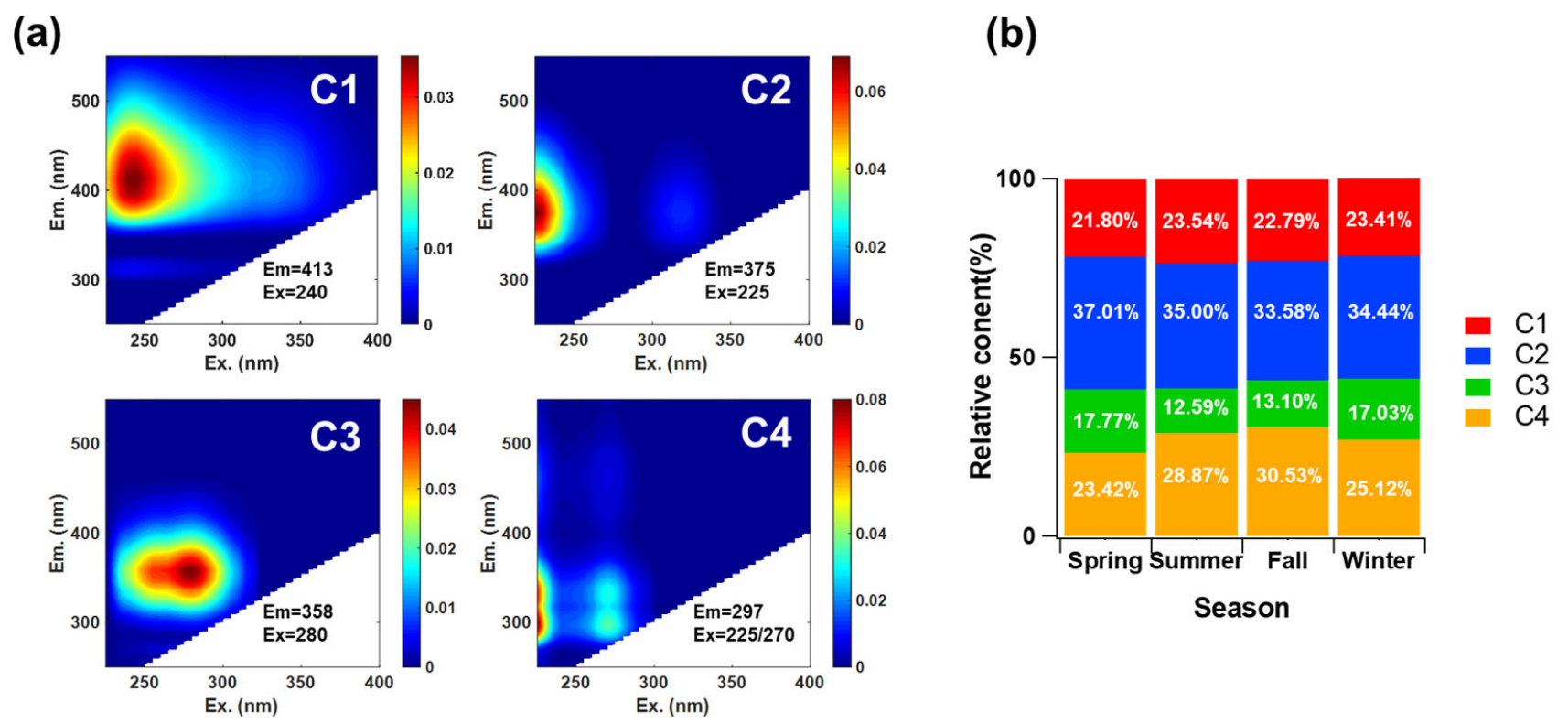

Figure 7 (a) Four EEM components identified by the PARAFAC model for the WSOA and (b) the relative contribution percentage of each component in different seasons.

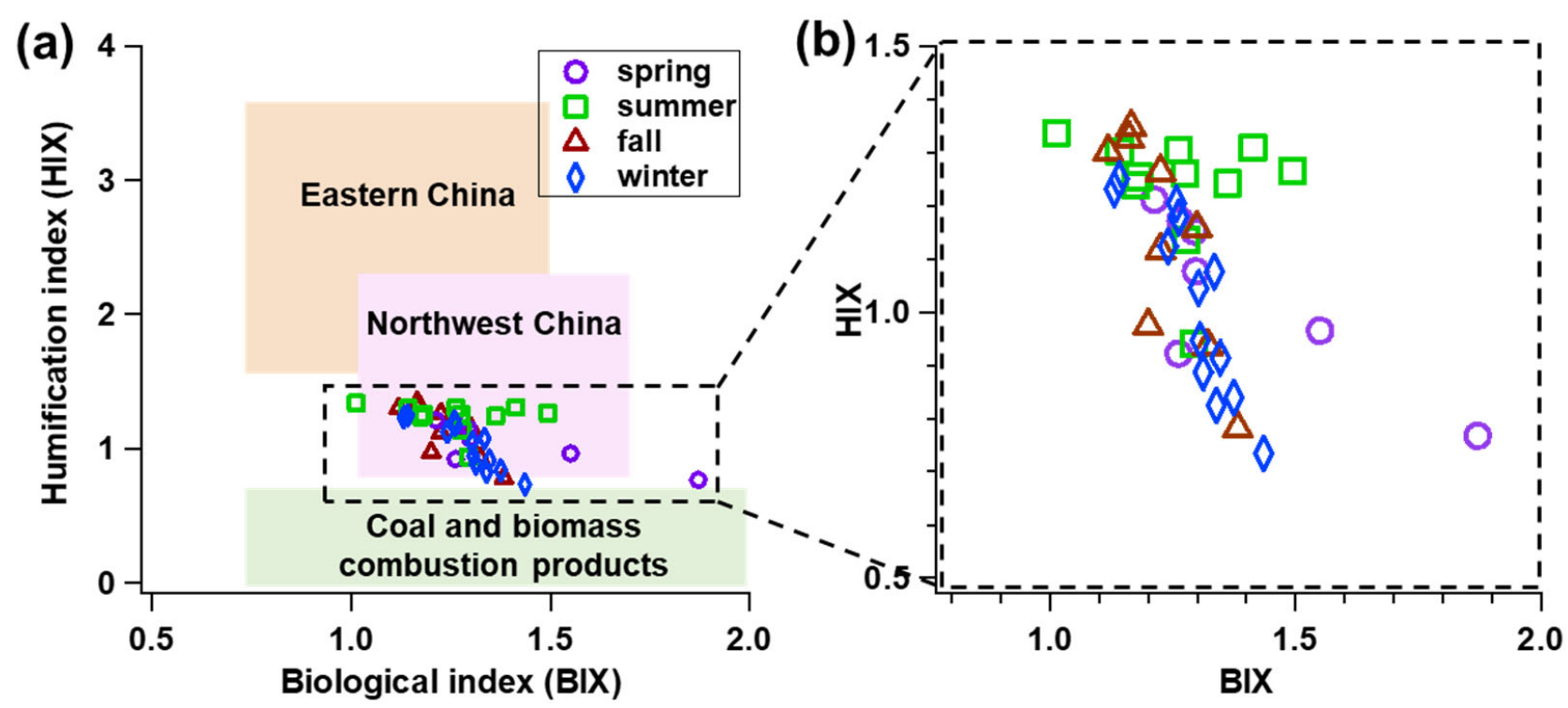

Figure 8 (a) and (b) BIX-HIX distribution map for four seasons, where orange, purple, and green boxes respectively represent the aerosol BIX-HIX range over eastern China, western China, and coal and biomass combustion products summarized by Zhong et al. (2023).

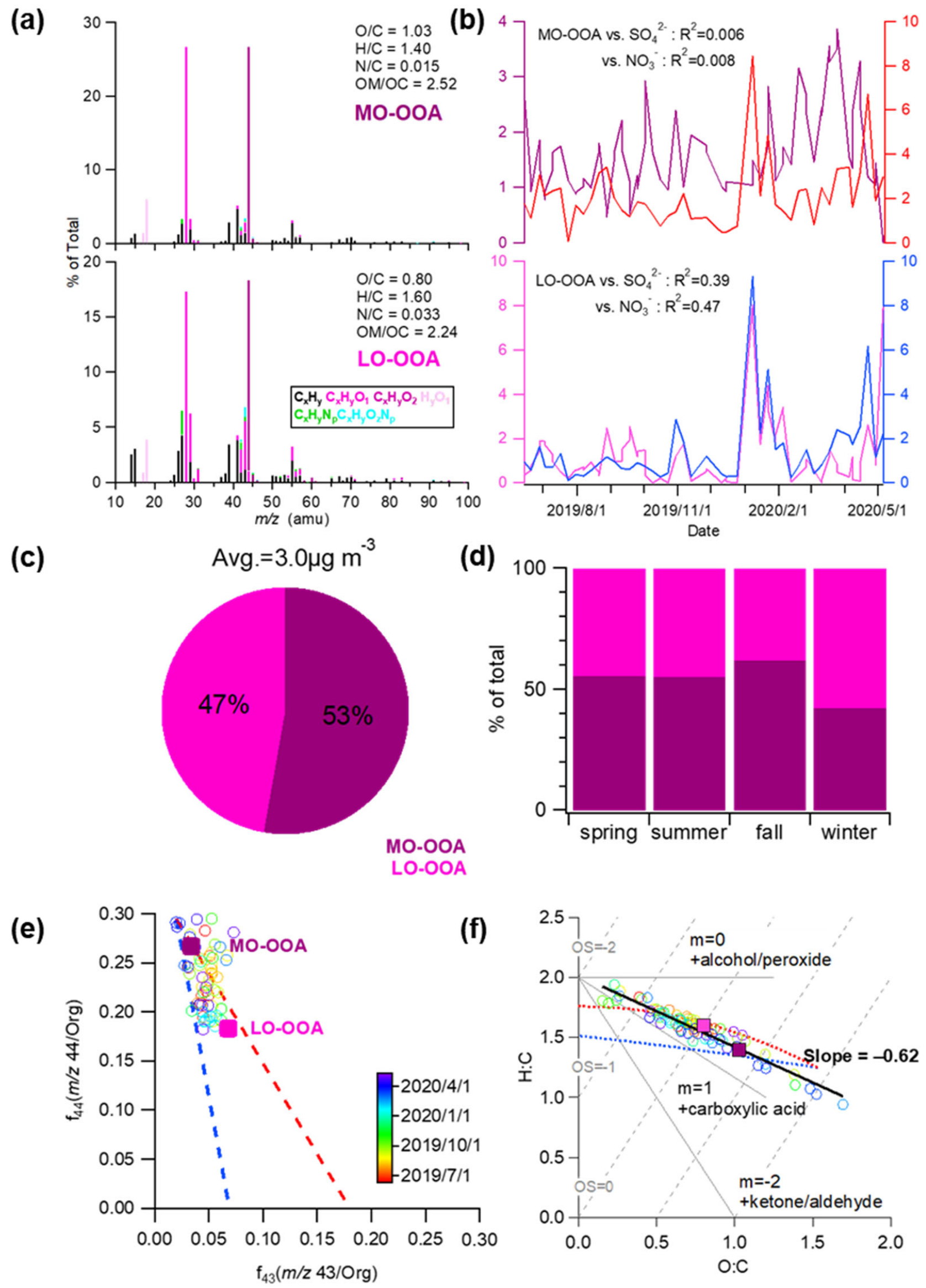

Figure 9 (a) PMF results of high-resolution mass spectra colored by six ion categories for the two OA factors at *m/z* < 120, (b) comparison of mass concentration time series changes of the two factors with their correlation of tracer species, (c) the average contribution of mass concentration of each factors to total organics, (d) the contribution percentage of two factors to the total mass in four seasons, (e) *f*44 vs. *f*43 triangle plot, and (f) the Van Krevelen diagram (H : C vs. O : C) for the WLG samples and OA components, where the red and blue dashed lines correspond to the same color dashed lines in the *f*44 vs. *f*43 triangle plot.

821 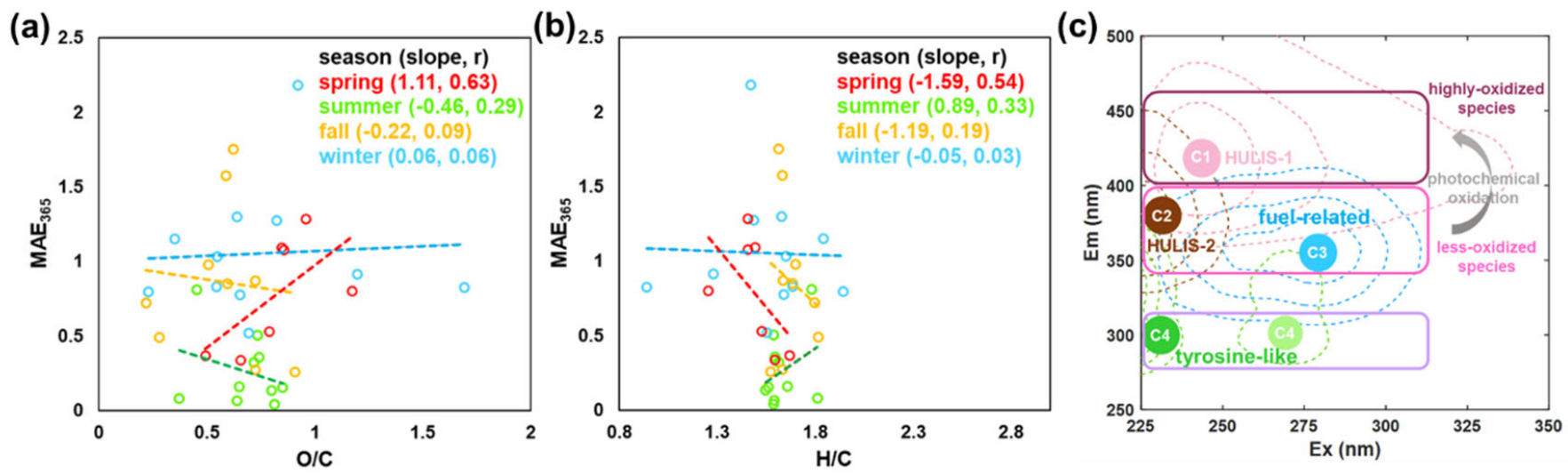

822 Figure 10 (a) and (b) Scatterplot of $MAE_{365}$ with O/C and H/C for four seasons. (c) The position of
823 fluorescence peak of chromophore and corresponding oxidizing species.

# Table

Table 1 Light-absorbing properties of BrC and fluorescence indices of WSOA in four seasons.

| **Season** | **$Abs_{365}$ (M/m)** | **$AAE_{300-400}$** | **$MAE_{365}$ ($m^2$/g)** | **HIX** | **BIX** |
|---|---|---|---|---|---|
| **Spring** | 1.45 ± 0.54 | 5.14 ± 1.46 | 0.92 ± 0.54 | 1.04±0.16 | 1.39±0.24 |
| **Summer** | 0.36 ± 0.21 | 6.21 ± 1.50 | 0.40 ± 0.24 | 1.24±0.11 | 1.26±0.13 |
| **Fall** | 0.88 ± 0.70 | 5.19 ± 1.00 | 0.81 ± 0.46 | 1.13±0.20 | 1.23±0.09 |
| **Winter** | 1.55 ± 1.30 | 5.48 ± 0.96 | 0.97 ± 0.49 | 1.02±0.17 | 1.29±0.09 |