# Peer review of "Measurement report: Year-long chemical composition,"

_EGUsphere, 2025_

## Referee Comment (RC1)

This manuscript provides a valuable year-long dataset from the Tibetan Plateau, including chemical composition and optical properties of aerosols. While the data coverage is extensive and multi-methodological (e.g., UV-Vis, PARAFAC, PMF), the manuscript often reads more like a report than a scientific discussion. Greater effort should be made to interpret the results in light of previous literature and physical processes. Below are detailed comments tied to specific line numbers for clarity.

-More introduction on the background studies in the region is needed. importance of one year long and integrated measurement studies are needed. Also need to discussion about the previous studies has been done in this area

**Methodology]**

- The method section heading should better reflect the full scope of the measurements (not just OC/EC or UV-Vis). Also need to be organized.

- Clarify how many samples were collected, and whether these are representative.

- Specify how OM, EC, and WSI were measured (e.g., AMS for OM? IC for WSI?) also need to explain how these data were used and what was the purpose for.

- OC does not necessarily represent the full water-soluble organic carbon — consider using WSOC instead.

**Line 282–286]**

- The claim that Ca2+ increases in spring is not strongly supported in the figure. Please refer to figures or provide clear quantitative evidence. — this undermines the linkage to dust transport.

- Sulfate peaks in summer, not spring — this undermines the linkage to anthropogenic influence.

- Wind direction discussion is unclear; winds in spring appear similar to other seasons — clarify or expand.

**Lines 307–309:**Oxalic acid $R^2$ = 0.4 is considered weak, not moderate. Revise the language and provide a better justification.

**Line 330:**The conclusion that low MAE365 in summer is due to photobleaching is speculative without any other supporting. Could it also be due to fewer chromophores? Further evidence is needed.

**Lines 333–336:** The MAE365 in spring is higher than at QOMS despite less biomass burning influence. Explain why this may be — secondary formation? Source region differences?

**Lines 344–349:** Comparing summer AAE with annual AAE from other studies is inconsistent unless justified. Align comparisons by season or explain the rationale.

**Lines 352–355:** The classification scheme from Saleh (2020) is introduced, but not clearly interpreted for your dataset. What does your MAE–AAE position tell us about the BrC types at this site?

**Line 362:** "Moderately absorptive BrC" implies some source or process — is this consistent with PMF or OA oxidation states?

**Lines 363–370, PARAFAC section:**The section lacks interpretation. What do the fluorescence numbers mean in terms of source or oxidation state? How does this support your conclusion?

**Line 399:** Only two OA factors (MO-OOA, LO-OOA) were resolved. Was there no biomass burning OA? Clarify if this was due to method limitation or absence of source.

**Line 417–420:** The slope analysis comparing O:C and H:C is unclear. What does a slope of –0.62 imply versus –0.89? Clarify the physical meaning.

**Lines 429–434:**Comparisons of MAC365 between urban and remote sites need qualification. The difference in formation pathways, oxidation levels, and sources must be considered.

**Lines 439–447:**The logic behind changes in MAE365 with O/C and H/C across seasons is not clearly supported. Summer O/C is not particularly high — so why the assumed aging effect?

**Lines 450–460:**The integration of PARAFAC and PMF components is potentially insightful but poorly explained. What are the identities and atmospheric implications of C1–C4? Why does C4 not correlate with OOA? Needs more in-depth discussion.

---

## Author Response (AR1)

**Measurement report: Year-long chemical composition, optical properties, and sources of atmospheric aerosols in the northeastern Tibetan Plateau**

**Response Letter to Reviewers' Comments**

Dear editors and reviewers,

We sincerely thank the editors and reviewers for their constructive comments and insightful suggestions. We have carefully revised the manuscript accordingly. A detailed, point-by-point response to each comment is provided below. For clarity, the reviewers' original comments are presented in *italic black*, our responses are in plain black, and the corresponding revisions in the manuscript are shown in blue. We hope that these revisions have adequately addressed all concerns and that the revised manuscript meets the expectations of the editors and reviewers.

**Reviewer #1:**

*This manuscript provides a valuable year-long dataset from the Tibetan Plateau, including chemical composition and optical properties of aerosols. While the data coverage is extensive and multi-methodological (e.g., UV-Vis, PARAFAC, PMF), the manuscript often reads more like a report than a scientific discussion. Greater effort should be made to interpret the results in light of previous literature and physical processes. Below are detailed comments tied to specific line numbers for clarity.*
*More introduction on the background studies in the region is needed. Importance of one year long and integrated measurement studies are needed. Also need to discussion about the previous studies have been done in this area.*

**Response:** We appreciate the reviewer's insightful comment. As this study is categorized as a "Measurement Report", our primary aim is to present substantial new results from measurements of atmospheric properties and chemical processes from field and laboratory experiments. We have improved the introduction section following the suggestion of the reviewer.

The background studies on BrC are included as follows:

"Recent studies on BrC have primarily concentrated on its sources, secondary formation pathways, optical properties, and radiative forcing effects (Ma et al., 2018; Chelluboyina et al., 2024)." (lines 50-52)

"Field and laboratory studies have revealed important transformations of BrC during atmospheric processes. For example, BrC can undergo photobleaching and oxidative whitening during long-range transport, leading to significantly reductions in its light absorption capacity (Sumlin et al., 2017). In addition, secondary BrC formed through photochemical reactions exhibit distinct diurnal

variability: enhanced light absorption in the morning due to active formation processes, followed by significant photobleaching under stronger oxidative conditions in the afternoon (Wang et al., 2019). Furthermore, recent work by Zhong et al. (2023) suggests that light absorption and fluorescence characteristics of BrC are influenced by environmental acidity, indicating the pH conditions may also modulate its optical behavior." (lines 54-63)

"In these marginal areas, BrC has been identified as a notable warming agent (Zhu et al., 2024). For example, BrC presented a higher absorption contribution compared to BC in the remote northeastern and southwestern margins of the TP (Zhu et al., 2021)." (lines 70-72)

"Orographic clouds dominate precipitation generation in the mountainous region (Qi et al., 2022) and aerosol-cloud interactions have become an increasing focus in the QLM (Liu et al., 2019; Xu et al., 2024)." (lines 76-78)"

*1. [Methodology] The method section heading should better reflect the full scope of the measurements (not just OC/EC or UV-Vis). Also need to be organized.*

**Response:** We appreciate the reviewer's suggestion. The headings for each part of the method section have been revised better to include more information on measurement and analysis conducted in this study. The content within each subsection has been reorganized and refined to enhance clarity, coherence, and focus.

"2.3.1 Measurements of carbonaceous materials
2.3.2 Ion chromatography analysis
2.3.3 UV-vis absorption measurements and analysis
2.3.4 EEM fluorescence spectra analysis
2.3.5 HR-ToF-AMS off-line measurement and PMF source decomposition
2.3.6 Source identification using backward trajectory model"

*2. [Methodology] Clarify how many samples were collected, and whether these are representative.*

**Response:** The sampling details are provided in Section 2.2: *"A total of 48 filter samples and three blank samples were collected during June 14, 2019 and May 6, 2020. Each sample was collected for 48h every seven days"*. The sampling schedule ensures a relatively uniform distribution across all four seasons. As shown in lines 307-310, the average mass concentrations of $PM_{2.5}$ exhibit clear seasonal variation that aligns with the regional climatic characteristics. Furthermore, the chemical composition of $PM_{2.5}$ also demonstrates seasonal differences, such as elevated contributions of dust-related components in spring and higher black carbon levels in winter. These patterns support the representativeness and reliability of our sampling strategy.

*3. [Methodology] Specify how OM, EC, and WSI were measured (e.g., AMS for OM? IC for WSI?) also need to explain how these data were used and what was the purpose for.*

**Response:** Sorry for not being clear about these components in our manuscript. The measurements

on these components are presented in section 2.3. We re-organize this section and add a few sentences to clarify the purpose of this measurement as follows.

"Our measurement strategy aimed to capture the chemical composition of $PM_{2.5}$ as comprehensively as possible. The chemical species were broadly categorized into inorganic and organic constituents. Inorganic components were quantified using IC, while organic species were determined based on organic carbon measurements. Additionally, the optical properties of organic species were characterized using UV-Vis and EEM spectroscopy. Detailed descriptions of each measurement are provided below." (lines 138-143)

*4. [Methodology] OC does not necessarily represent the full water-soluble organic carbon — consider using WSOC instead.*

**Response:** In the updated manuscript, we have carefully distinguished between "organic carbon (OC)" and "water-soluble organic carbon (WSOC)". Specifically, OC refers to the total organic carbon content as measured and described in Section 2.3.1, whereas WSOC represents the fraction of OC that is soluble in water. We have thoroughly reviewed the manuscript to ensure that these terms are used consistently and accurately throughout.

*5. [Lines 282–286] The claim that $Ca^{2+}$ increases in spring is not strongly supported in the figure. Please refer to figures or provide clear quantitative evidence. — this undermines the linkage to dust transport.*

**Response:** The original manuscript presented the impact of mineral dust during spring through the seasonal variation in the percentage of $Ca^{2+}$ showing a **peak in spring (7.5% of $Ca^{2+}$) and a minimum in summer (4.1% of $Ca^{2+}$).**. However, the percentage differences may not be sufficiently prominent to reflect the seasonal dust contribution. To address this, we have revised the presentation by using the absolute concentration of $Ca^{2+}$ to better characterize seasonal changes.

"The natural mineral dust reached its peak in spring (1.05 $\mu$g m$^{-3}$ of $Ca^{2+}$) and its minimum in summer (0.29 $\mu$g m$^{-3}$ of $Ca^{2+}$) (Figure 2c)." (lines 296-297)

*6. [Lines 282–286] Sulfate peaks in summer, not spring — this undermines the linkage to anthropogenic influence.*

**Response:** We agree with the reviewer's comment regarding the peak sulfate contribution in summer, which is closely associated with enhanced anthropogenic emissions and intensified secondary aerosol formation. The summer peak of sulfate is indeed a key feature in this region and is consistent with previous findings in the Tibetan Plateau. In our study, $SO_4^{2-}$ and $NO_3^-$ were selected as markers of anthropogenic pollution, as their gaseous precursors — $SO_2$ and $NO_X$ — are commonly emitted from anthropogenic sources. Moreover, we observed strong correlations between $SO_4^{2-}$ and $NO_3^-$ ($R^2$=0.90), as well as between $SO_4^{2-}$ and $NH_4^+$ ($R^2$=0.91), indicating that sulfate is primarily formed through secondary oxidation processes. The summer season provides favorable conditions for these reaction conditions, leading to elevated sulfate contribution. It is

important to note that this enhancement is not primarily due to increased emissions, but rather due to more active atmospheric oxidation and secondary formation mechanisms. These processes and their seasonal characteristics have been further elaborated upon in the revised manuscript.

*7. [Lines 282–286] Wind direction discussion is unclear; winds in spring appear similar to other seasons — clarify or expand.*

**Response:** The description of seasonal wind patterns has been clarified in the first paragraph of Section 3, based on the presentation of Figure 2a. Specifically, wind directions were predominantly from the west during winter, with a step increase from the east during spring. This east wind reached the maximum in summer, reflecting a transition from the westerlies to summer monsoon systems. Fall represented a transitional period.

*8. [Lines 307–309] Oxalic acid $R^2 = 0.4$ is considered weak, not moderate. Revise the language and provide a better justification.*

**Response:** For oxalic acid, although the determination coefficient ($R^2 = 0.40$) appears modest, the corresponding Pearson's r is 0.62, which suggests a moderate positive relationship. Moreover, the correlation is statistically significant based on t-test ($p < 0.05$). We have updated the manuscript to clarify both the strength and statistical significance of this relationship.

*9. [Line 330] The conclusion that low $MAE_{365}$ in summer is due to photobleaching is speculative without any other supporting. Could it also be due to fewer chromophores? Further evidence is needed.*

**Response:** We agree that a low $MAE_{365}$ value may result from either a reduced abundance of light-absorbing chromophores or photobleaching effects. In the revised manuscript, we have clarified that both factors could contribute to the observed seasonal variation. However, we propose that photobleaching plays a predominant role, based on the observed negative correlation between $MAE_{365}$ and the oxidation state of OA. This relationship suggests that more highly oxidized OAs are associated with lower light absorption efficiency, consistent with photobleaching effect.

*10. [Lines 333–336] The $MAE_{365}$ in spring is higher than at QOMS despite less biomass burning influence. Explain why this may be — secondary formation? Source region differences?*

**Response:** The higher $MAE_{365}$ observed in spring at WLG, despite a lower influence from biomass burning compared to QOMS, may be attributed to stronger regional anthropogenic emissions around the WLG site. This is supported by the higher PM mass concentrations observed at WLG relative to QOMS during the same season. At QOMS, the air masses typically undergo long-range transport with extensive atmospheric oxidation, leading to more aged OA with higher oxygen-to-carbon (O/C) ratio and reduced light-absorbing capacity. In contrast, air masses arriving at WLG during spring were often transported over shorter distances from surrounding urban areas, carrying more primary OA that contain stronger light-absorbing chromophores. These differences in source distance and aging processes likely contribute to the higher $MAE_{365}$ observed at WLG.

*11. [Lines 344–349] Comparing summer AAE with annual AAE from other studies is inconsistent unless justified. Align comparisons by season or explain the rationale.*

**Response:** We agree that comparisons of AAE values should be made using data from the same season or averaged annually to ensure consistency. In the revised manuscript, we have updated the comparisons accordingly by aligning them either seasonally or annually, depending on the availability of data in the referenced studies as follows:

"The average annual AAE is comparable with the observation at Lulang (5.39 ± 1.22, 330 – 400 nm), in the southeastern part of the TP, during August 2014 to August 2015 (Li et al., 2016b) and Lhasa (5.38, 330 – 400 nm) during May 2013 to March 2014, a typical urban area on the TP (Li et al., 2016a). The summertime AAE is also similar to those at other stations on the TP, such as Nam Co (5.91 ± 2.14, 300 – 400 nm) from May 13 to July 1, 2015 (Zhang et al., 2017) and WLG (5.96, 300 – 400 nm) from July 2017 (Xu et al., 2020). The AAE value in winter is closed to that observed in the eastern Himalayas (5.5, 300 – 450 nm) during the 2019-2020 winter (Arun et al., 2024)." (lines 359-365)

*12. [Lines 352–355] The classification scheme from Saleh (2020) is introduced, but not clearly interpreted for your dataset. What does your MAE–AAE position tell us about the BrC types at this site?*

**Response:** In our dataset, the majority of data points in the MAE-AAE space fall within the region corresponding to weakly absorbing brown carbon (W-BrC). This suggests that the OA at WLG were overall highly oxidized, which is consistent with observations from other remote sites on the Tibetan Plateau. A limited number of data points, primarily from the spring, deviate from this pattern and fall closer to the region associated with moderately absorbing BrC (M-BrC). These outliers were likely influenced by episodic inputs of anthropogenic aerosols from surrounding populated areas. We have updated the manuscript to include this interpretation and to clarify the relevance of the MAE-AAE space at our site.

*13. [Line 362] "Moderately absorptive BrC" implies some source or process — is this consistent with PMF or OA oxidation states?*

**Response:** The presence of moderately absorptive BrC is consistent with the PMF results or the observed oxidation state of OA. As mentioned in our responses to comments 10-12, several samples collected during spring exhibited relatively lower O/C ratios, yet they were still identified as SOA. For these spring samples, we observed a positive correlation between the light absorption capacity and the oxidation state of OA. This suggests that secondary formation such as functionalization processes can enhance light-absorbing properties. Therefore, the occurrence of moderately absorptive BrC during spring is consistent with both the aerosol source apportionment and the chemical evolution pathways reflected in our data. We have clarified this interpretation in the revised manuscript.

*14. [Lines 363–370, PARAFAC section:] The section lacks interpretation. What do the fluorescence numbers mean in terms of source or oxidation state? How does this support your conclusion?*

**Response:** The four fluorescence components identified through PARAFAC analysis, labeled as C1, C2, C3, and C4, are in line with common practice in previous studies. The excitation-emission (Ex/Em) spectra of each component were analyzed to infer their chemical characteristics and potential sources. For example, components with humic-like (C1 and C2) or protein-like (C3 and C4) fluorescence signatures were linked to terrestrial or microbial origins, respectively, and their spectral features were used to assess their degree of oxidation. Furthermore, in Section 3.5, we examined the relationships between these fluorescence components and the AMS-derived chemical parameters using EEM-plot (Figure 10c). This analysis supports our conclusion by demonstrating how specific fluorescent components are associated with SOA formation and oxidation state.

*15. [Line 399] Only two OA factors (MO-OOA, LO-OOA) were resolved. Was there no biomass burning OA? Clarify if this was due to method limitation or absence of source.*

**Response:** In our study, only two OA factors were decomposed by PMF analysis on mass spectra of WSOA. This result does not necessarily indicate the complete absence of primary sources such as BBOA and HOA, but rather reflects methodological limitation. Since our analysis was based solely on the water-soluble fraction of OA, hydrophobic primary components may not be captured effectively in the PMF decomposition. These components may still have been present, particularly during winter and spring. We have added this explanation to the revised manuscript to clarify the scope and limitations of our PMF results.

*16. [Line 417–420] The slope analysis comparing O:C and H:C is unclear. What does a slope of –0.62 imply versus –0.89? Clarify the physical meaning.*

**Response:** The slope in the Van Krevelen diagram (H:C vs. O:C) reflects the dominant oxidation pathways of OA. As shown in Ng et al. (2011), a slope of –1.0 typically indicates the addition of carboxylic acid functional groups, while a slope of 0 implies the addition of alcohol or peroxide groups, which increase the oxygen content without significant hydrogen loss. In our study, the slope of –0.62 suggests that the SOA at our site is formed through a mixed oxidation mechanism, involving both carboxylic acid and alcohol/peroxide formation pathways. This interpretation has now been clarified and updated in the revised manuscript as follows.

"The slope of the linear regression in the diagram reflects the predominant oxidation pathways. A slope between –1 to –0.5 is typically associated with functionalization processes such as the formation of carboxylic acids or alcohol/peroxide groups (Ng et al., 2011). In our dataset, the overall slope was –0.62 (Figure 9f), indicating mixed contributions from both carboxylic acid and alcohol/peroxide formation pathways. This slope was higher than the value previously reposted for winter in Xining (–0.89) and summer at NamCo (–0.76) (Xu et al., 2018; Zhong et al., 2023)." (lines 439-445)

*17. [Lines 429–434] Comparisons of MAC$_{365}$ between urban and remote sites need qualification.*

*The difference in formation pathways, oxidation levels, and sources must be considered.*

**Response:** We agree with the reviewer that comparisons of $MAC_{365}$ of OOA components across different sites should take into account differences in formation pathways, oxidation states, and source characteristics. In fact, one of the objectives of presenting such comparisons is to highlight how these factors influence the variability in $MAC_{365}$ values. We acknowledge that the chemical aging and source profiles differ substantially between urban, rural, and remote sites, and these differences are reflected in the absorptive properties of OA. Furthermore, studies that report MAC values specifically for PMF-resolved OOA components are still limited in the literature. Therefore, although the comparisons require qualification, they can still provide useful context and insight into the role of atmospheric processing in BrC evolution. In the revised manuscript, we have added additional explanation to clarify the rationale behind these comparisons and to better account for the underlying differences among the sites.

*18. [Lines 439–447] The logic behind changes in $MAE_{365}$ with O/C and H/C across seasons is not clearly supported. Summer O/C is not particularly high — so why the assumed aging effect?*

**Response:** The interpretation of seasonal variations in $MAE_{365}$ with respect to O/C and H/C ratios is based on the results of Jiang et al. (2022) and Zhong et al. (2023). These two studies found that atmospheric aging of OA generally follows two stages. In the initial stage, functionalization dominates, wherein oxygen-containing functional groups are added to the OA molecules. This process typically enhances chromophore abundance and light absorption, leading to a positive correlation between MAE and O/C. However, with further atmospheric aging, oxidative fragmentation becomes dominant, producing highly oxidized, smaller, and more polar compounds. These products tend to lack strong chromophores and are more prone to photobleaching, resulting in a negative correlation between MAE and O/C.

In our study, although the average O/C value in summer is not the highest among all seasons, we observe that summer OA exhibits characteristics of extensive aging — such as relatively low $MAE_{365}$ and high degrees of oxidation — suggesting it has undergone significant oxidative processing and chromophore degradation. Therefore, we infer that the lower $MAE_{365}$ observed in summer is primarily due to the second stage aging effects, particularly photobleaching and fragmentation. We have clarified this reasoning in the revised manuscript accordingly.

"A low $MAE_{365}$ value may result from either a reduced abundance of light-absorbing chromophores or photobleaching effects. However, we propose that photobleaching plays a predominant role, based on the observed negative correlation between $MAE_{365}$ and the oxidation state of OA (see section 3.5)." (lines 342-345)

*19. [Lines 450–460] The integration of PARAFAC and PMF components is potentially insightful but poorly explained. What are the identities and atmospheric implications of C1–C4? Why does C4 not correlate with OOA? Needs more in-depth discussion.*

**Response:** This section of the manuscript aims to explore the linkage between the oxidation state

of OA and their fluorescence characteristics. The identifies and possible sources of the four PARAFAC components (C1-C4) are described in detail in Section 3.3. Specifically, C1 and C2 are associated with HULIS, which are typically related to aged or highly processed secondary organic matter. In contrast, C4 is characterized by a tyrosine-like fluorescence signature and is more influenced by primary emissions, including those from agricultural and ecological sources.

In Section 3.5, we discuss the correlations between these components and the PMF-resolved OA factors (MO-OOA and LO-OOA). Among them, C4 shows only a weak correlation with both OOA factors, suggesting that it is less influenced by oxidative processing and more reflective of direct emissions. This explains why C4 does not co-vary with the oxidation-related OA components and thus does not contribute significantly to the aging processes observed in this study.

We have revised and expanded the relevant discussion in the manuscript to clarify the identities and implications of the PARAFAC components.

"In this study, cross-correlation analysis was performed on the chemical components of $PM_{2.5}$ ($SO_4^{2-}$, $NO_3^-$, $NH_4^+$, $K^+$, $Cl^-$, LO-OOA, MO-OOA), oxidation degree (O/C and H/C), and the four PARAFAC-derived fluorescent components (C1 – C4). As illustrated in Figure S1, C1, C2, and C3 exhibited positive correlations with the secondary species ($SO_4^{2-}$, $NO_3^-$, $NH_4^+$, $Cl^-$, LO-OOA, and MO-OOA). The strongest correlations were observed between C1 and these species, suggesting a secondary source origin for this chromophore. In contrast, C4 showed weak or no significant correlations with any of the chemical components, implying that C4 was not directly related to the oxidation processes of OA. As discussed in Section 3.3, C4 was related to agriculture emissions and ecological activities, rather than secondary atmospheric formation. To further investigate the optical evolution of WS-BrC during the atmospheric oxidation processing, the relationships of PARAFAC components and the WSOA components were integrated in EEM plot (Figure 10c). C1 was associated with MO-OOA, whereas C2 and C3 were linked to LO-OOA, and C4 exhibited weak correlations with these two factors. This classification enables cross-validation of chemical and optical properties, providing additional insights into the formation pathways of chromophore components (Chen et al., 2016; Zhong et al., 2023). Overall, the chemical transformation from less oxidized to highly oxidized OA through photochemical reactions can be extended to the process of BrC. Correspondingly, the optical evolution of BrC can serve as evidence of the oxidative state transition." (lines 477-494)

[Figure]

Figure S1 Correlation heat map of chemical components of $PM_{2.5}$ ($SO_4^{2-}$, $NO_3^-$, $NH_4^+$, $K^+$, $Cl^-$, LO-OOA, MO-OOA), oxidation degree (O/C and H/C), and the four PARAFAC-derived fluorescent components (C1 – C4). The correlation coefficient and the results of the significance test are included.

**Reviewer #2:**

*This study reports aerosol characteristics from a remote site in the Tibetan Plateau. The authors find seasonal variation in the aerosol composition and absorption properties. The analysis is strengthened by the use of multiple complimentary analysis techniques, providing results of interest to the community.*

**Response:** We sincerely appreciate the reviewer's positive and encouraging comments.

*1. Throughout the manuscript, the authors note that many of the seasonal differences in the absorption properties are due to photobleaching (i.e. line 330, line 447). However, it is also possible that the aerosol sources are different between the seasons. For example, you mention coal burning as an aerosol source in the winter, which is highly absorbing. Can you comment on if you do expect different sources across seasons at this site? Additionally, do you expect a role of non-absorbing SOA (such as biogenic SOA) to decrease the MAE in the summer.*

**Response:** We agree that the different chemical composition and sources during each season could contribute to the difference of the seasonal MAE, especially for that between summer and winter. A more detailed explanations on these absorption differences are provided in the revised manuscript as follows.

"A low $MAE_{365}$ value may result from either a reduced abundance of light-absorbing chromophores or photobleaching effects. However, we propose that photobleaching plays a predominant role, based on the observed negative correlation between $MAE_{365}$ and the oxidation state of OA (see section 3.5)." (lines 342-345)

*2. It would be useful to add additional information on how to interpret the HIX and BIX results in section 3.3. It is clear that there are seasonal differences, however, further information on the importance of the differences would be useful.*

**Response:** In the revised manuscript, we have expanded the interpretation of the HIX and BIX results to better illustrate the seasonal differences and their atmospheric implications.

"The fluorescence indices BIX and HIX serve as complementary tools for characterizing fluorescent OM. BIX is particularly used to assess the biological freshness of OM, whereas HIX reflects the degree of humification and chemical aging (Lee et al., 2013). By integrating these two indices, a more comprehensive understanding of the properties of OM can be achieved. An elevated degree of aging is associated with increased HIX values (Fan et al., 2020; Wu et al., 2021; Ma et al., 2022) and decreased BIX values (Wen et al., 2021). In this study, the annual average HIX and BIX values were $1.11 \pm 0.18$ and $1.29 \pm 0.14$, respectively, with seasonal variations of $1.04 \pm 0.16$ and $1.39 \pm 0.24$ in spring, $1.24 \pm 0.11$ and $1.26 \pm 0.13$ in summer, $1.13 \pm 0.20$ and $1.23 \pm 0.09$ in fall, and $1.02 \pm 0.17$ and $1.29 \pm 0.09$ in winter. The spring samples exhibited the greatest variability, indicating their relatively fresh properties (Figure 8b). In contrast, summer was characterized by the highest HIX, suggesting a high degree of aging and oxidation. The HIX of the autumn samples were at an intermediate level, while winter samples had the lowest HIX and moderate BIX values, indicating the lower degree of oxidation (Figure 8a). Compared with previous studies, the fluorescence properties of aerosols in this study are more consistent with those in the northwestern China, and were less humified than those in the eastern China (Figure 8) (Chen et al., 2021; Zhang et al., 2021; Zhong et al., 2023)." (lines 401-416)

*3. Sections 2.3.3 and 2.3.4: These sections should be renamed to something more informative.*

**Response:** We agree with the reviewer's suggestion. The titles of sub-sections of the method section have been revised to more accurately reflect their content. Specifically, the updated titles are as follows:

"2.3.3 UV-vis absorption measurements and analysis"
"2.3.4 EEM fluorescence spectra analysis"

*4. [Line 206] Should this be nondispersive infrared gas detector?*

**Response:** Yes. We have revised this typo. In the updated manuscript, this sub-section has been merged into Section 2.3.1.

*5. [Line 226] This sentence is vague, was PMF run using PET2?*

**Response:** This sentence has been improved as follows:

". The data is processed using standard ToF-AMS data analysis toolkits SQUIRREL (v1.56) and PIKA (v1.15c) within Igor Pro 6.37. The processed matrix data were employed to investigate the sources of WSOA by positive matrix factorization (PMF). PMF source analysis was processed using the standard PMF evaluation tool (PET v2.03) developed based on the PMF2.exe algorithm (Ulbrich et al., 2009)." (lines 237-241)

*6. [Section 2.3.7] How often were trajectories initiated?*

**Response:** In our study, 72-hour backward trajectories were initiated on an hourly frequency throughout the sampling period. This high temporal resolution was chosen to capture the fine-scale variability in air mass origins and transport pathways.

*7. [Line 270] What are the percentage values for precipitation? Percent of days with precipitation?*

**Response:** The results are presented at the frequency and updated the revised manuscript.

*8. [Line 283] It doesn't seem that $Ca^{2+}$ is that much higher in spring than fall, and it is not negligible even in summer and winter. Is there other data to support increased mineral dust in spring, or is mineral dust present year round.*

**Response:** We agree with the reviewer that the relative contribution of $Ca^{2+}$ does not show a drastic seasonal variation, likely due to the persistently semiarid environment surrounding the sampling site. However, both the average concentration and standard deviation of $Ca^{2+}$ are notably higher in spring compared to other seasons. Specifically, the seasonal mean concentrations of $Ca^{2+}$ were 1.05 µg m$^{-3}$ in spring, 0.29 µg m$^{-3}$ in summer, 0.54 µg m$^{-3}$ in autumn, and 0.56 µg m$^{-3}$ in winter. These values indicate a substantial increase in $Ca^{2+}$ concentrations during spring, suggesting an enhanced input of mineral dust in that season. To better illustrate this point, we have revised the manuscript to include both the seasonal average concentrations and their respective contributions.

*9. Consider expanding the discussion in the paragraph starting at line 449. The combination of the EEM and PMF data is intriguing, however, I'm having trouble understanding this paragraph. Can you provide details on what is meant by "correlation analysis"? Why is there no correlation seen with C4?*

**Response:** Thank you for your valuable comments. We apologize for the lack of clarification in the original manuscript. The purpose of combining EEM and PMF data is to link the optical properties of WSOA with their chemical composition. This is achieved through correlation analysis performed across all identified EEM and PMF factors, as well as relevant inorganic species. Specifically, we calculate correlation coefficients to quantitatively assess the relationships between these factors (Figure S1). Based on the correlation results, the corresponding factors are then visualized on the EEM plot to facilitate interpretation. Regarding the C4 factor, it is characterized as protein-like

material, which likely originates from ecological emissions rather than atmospheric chemical processes. Therefore, it does not exhibit significant correlations with the PMF factors derived from atmospheric chemistry. We have revised the manuscript to clarify these points for better understanding.

[Figure]

Figure S1 Correlation heat map of chemical components of $PM_{2.5}$ ($SO_4^{2-}$, $NO_3^-$, $NH_4^+$, $K^+$, $Cl^-$, LO-OOA, MO-OOA), oxidation degree (O/C and H/C), and the four PARAFAC-derived fluorescent components (C1 – C4). The correlation coefficient and the results of the significance test are included.

*10. [Figure 2] It is difficult to see the time trends in the minor components due to the scale. Consider revising the figure to better show all the species, for example splitting the middle panel into two axes.*

**Response:** We thank the reviewer for this constructive suggestion. Due to the high concentration of some samples in winter and spring, this phenomenon is inevitable. We adjusted the height of the vertical axis and the figure to make some minor components visible more clearly.

*There are several grammatical errors throughout the manuscript. I recommend the authors carefully proofread the manuscript. A few specific cases are listed below.*

**Response:** We sincerely apologize for the grammatical oversights and deeply appreciate the reviewer's meticulous attention to language quality. We have revised the manuscript according to the suggestions given.

*11. [Line 41] "···and strongly interacted with ambient conditions during transport" meaning of this sentence is unclear*

**Response:** The revised manuscript now explicitly defines the physicochemical processes underlying aerosol-environment interactions during transport.

"During atmospheric transport, aerosol undergo extensive physicochemical transformations driven by environmental factors such as relative humidity, oxidants, and solar radiation (Lee et al., 2008; Chen and Torres, 2009; Yu et al., 2022; Klodt et al., 2023)." (lines 39-41)

*12. [Line 65] "Precipitation in the mountain areas, through aerosol-cloud interaction, is the major origination (Qi et al., 2022)" meaning is unclear.*

**Response:** The revised text now clearly clarifies the influence of topographic clouds on precipitation in the QLM region and leads to the discussion of aerosol-cloud interactions.

"Orographic clouds dominate precipitation generation in the mountainous region (Qi et al., 2022) and aerosol-cloud interactions have become an increasing focus in the QLM (Liu et al., 2019; Xu et al., 2024)." (lines 76-78)

*13. [Line 278] $NH^{4+}$ should be $NH_4^+$.*

**Response:** Corrected.

**References**

[revised manuscript text omitted]

---

## Author Response (AR2)

**Atmospheric Chemistry and Physics**
**Ref: EGUSPHERE-2025-41**

**Measurement report: Year-long chemical composition, optical properties, and sources of atmospheric aerosols in the northeastern Tibetan Plateau**

**Response Letter to Reviewers' Comments**

Dear editors and reviewers,

We sincerely thank the editors and reviewers for their constructive comments and insightful suggestions. We have carefully revised the manuscript accordingly. A detailed, point-by-point response to each comment is provided below. For clarity, the reviewers' original comments are presented in *italic black*, our responses are in red, and the corresponding revisions in the manuscript are shown in blue. We hope that these revisions have adequately addressed all concerns and that the revised manuscript meets the expectations of the editors and reviewers.

**Response to Reviewer #3:**

*This is a revised paper after addressing the reviewers' comments. After a careful reading of the original paper and the responses to the reviewers as well as the revised version, this reviewer think the authors have relatively well addressed the issues raised by the reviewer, and the paper can be accepted by the ACP. I only have a few minor suggestions*

*(1) In order to make the seasonal differences more robust and reliable, I suggest to do a student t-test also on the different pairs of Abs365 data and provide the p values.*

**Response:** As suggested by the reviewer, we conduct pairwise Student's t-tests to evaluate the statistical significance of seasonal variations in $Abs_{365}$, $MAE_{365}$ and $PM_{2.5}$ mass concentrations. The results indicate that the most pronounced seasonal difference in $Abs_{365}$ occurs between spring and summer ($p < 0.01$), followed by summer-winter and fall-spring ($p < 0.1$). In contrast, the differences between winter-spring, fall-winter, and summer-fall are not statistically significant ($p > 0.1$).

$MAE_{365}$ exhibits a statistically significant difference between summer and winter (p < 0.05) and a marginally significant difference between spring and autumn (p < 0.1). Furthermore, $PM_{2.5}$ concentrations between spring and winter are significantly higher than those in summer and autumn ($p < 0.05$); while the difference between spring and winter, as well as between summer and autumn are not significant ($p > 0.1$). These results have been incorporated into the revised manuscript accordingly.

"Seasonal Student's t-tests revealed that the mass concentrations during spring (14.0 µg m$^{-3}$) and winter (12.5 µg m$^{-3}$) were both significantly higher than those in summer (7.1 µg m$^{-3}$) and fall (8.0 µg m$^{-3}$) ($p < 0.05$) (Figure 2c)" (lines 293-295)

"The results of the Student's t-test confirmed that the most pronounced seasonal difference in $Abs_{365}$ occurred between spring and summer ($p < 0.01$), followed by the pairs of summer-winter and fall-spring ($p < 0.1$)." (lines 339-341)

"The average absorption efficiency of WS-BrC at unit WSOC content (MAE) during the summer at 365nm ($MAE_{365}$, $0.40 \pm 0.24$ $m^2g^{-1}$) is significantly lower than that of the other three seasons ($0.92 \pm 0.54$ $m^2g^{-1}$ in spring, $0.81 \pm 0.46$ $m^2g^{-1}$ in fall and $0.97 \pm 0.49$ $m^2g^{-1}$ in winter) ($p < 0.1$) (Figure 6a)." (lines 343-347)

*(2) As your measured OC and also water-soluble OC, but the source analysis was only conducted on WSOC which is understandable due to technical limitations. Can you provide a scatter plot of WSOC vs OC, and check the correlations and slope, therefore readers can know the recovery ratio of WSOC from OC, and understand the representativeness of WSOC for OC.*

**Response:** As suggested by the reviewer, we examine the relationship between WSOC and OC to evaluate the representativeness of WSOC for the total OC. The WSOC/OC values range from 0.31 to 0.99, with an average of $0.63 \pm 0.21$. A strong positive correlation is observed between WSOC and OC ($R^2 = 0.81$) (Fig. S1). These results indicate that WSOC accounts for a substantial fraction of OC and can serve as a reliable representativeness for total OC. In the updated manuscript, relevant descriptions have all been included as follows.

"Note that the relationship analysis between WSOC and OC indeed exhibited a tight correlation ($R^2 = 0.81$) and a relatively high ratio ($0.63 \pm 0.21$), suggesting the representativeness of water-soluble fraction on bulk OM." (lines 341-343)

[Figure]

Figure S1 Scatter plot of WSOC and OC.